# On Quadratic Convergence of DC Proximal Newton Algorithm in Nonconvex Sparse Learning

**Xingguo Li**[1,4]    **Lin F. Yang**[2*]    **Jason Ge**[2]    **Jarvis Haupt**[1]    **Tong Zhang**[3]    **Tuo Zhao**[4†]
[1]University of Minnesota   [2]Princeton University   [3]Tencent AI Lab   [4]Georgia Tech

## Abstract

We propose a DC proximal Newton algorithm for solving nonconvex regularized sparse learning problems in high dimensions. Our proposed algorithm integrates the proximal newton algorithm with multi-stage convex relaxation based on the difference of convex (DC) programming, and enjoys both strong computational and statistical guarantees. Specifically, by leveraging a sophisticated characterization of sparse modeling structures (i.e., local restricted strong convexity and Hessian smoothness), we prove that within each stage of convex relaxation, our proposed algorithm achieves (local) quadratic convergence, and eventually obtains a sparse approximate local optimum with optimal statistical properties after only a few convex relaxations. Numerical experiments are provided to support our theory.

## 1   Introduction

We consider a high dimensional regression or classification problem: Given $n$ independent observations $\{x_i, y_i\}_{i=1}^n \subset \mathbb{R}^d \times \mathbb{R}$ sampled from a joint distribution $\mathcal{D}(X, Y)$, we are interested in learning the conditional distribution $\mathbb{P}(Y|X)$ from the data. A popular modeling approach is the Generalized Linear Model (GLM) [20], which assumes

$$\mathbb{P}\left(Y|X; \theta^*\right) \propto \exp\left(\frac{YX^\top\theta^* - \psi(X^\top\theta^*)}{c(\sigma)}\right),$$

where $c(\sigma)$ is a scaling parameter, and $\psi$ is the cumulant function. A natural approach to estimate $\theta^*$ is the Maximum Likelihood Estimation (MLE) [25], which essentially minimizes the negative log-likelihood of the data given parameters. However, MLE often performs poorly in parameter estimation in high dimensions due to the curse of dimensionality [6].

To address this issue, machine learning researchers and statisticians follow Occam's razor principle, and propose sparse modeling approaches [3, 26, 30, 32]. These sparse modeling approaches assume that $\theta^*$ is a sparse vector with only $s^*$ nonzero entries, where $s^* < n \ll d$. This implies that many variables in $X$ are essentially irrelevant to modeling, which is very natural to many real world applications such as genomics and medical imaging [7, 21]. Many empirical results have corroborated the success of sparse modeling in high dimensions. Specifically, many sparse modeling approaches obtain a sparse estimator of $\theta^*$ by solving the following regularized optimization problem,

$$\overline{\theta} = \underset{\theta \in \mathbb{R}^d}{\operatorname{argmin}} \mathcal{L}(\theta) + \mathcal{R}_{\lambda_{\mathrm{tgt}}}(\theta), \tag{1}$$

where $\mathcal{L} : \mathbb{R}^d \to \mathbb{R}$ is the convex negative log-likelihood (or pseudo-likelihood) function, $\mathcal{R}_{\lambda_{\mathrm{tgt}}} : \mathbb{R}^d \to \mathbb{R}$ is a sparsity-inducing decomposable regularizer, i.e., $\mathcal{R}_{\lambda_{\mathrm{tgt}}}(\theta) = \sum_{j=1}^d r_{\lambda_{\mathrm{tgt}}}(\theta_j)$ with $r_{\lambda_{\mathrm{tgt}}} : \mathbb{R} \to \mathbb{R}$, and $\lambda_{\mathrm{tgt}} > 0$ is the regularization parameter. Many existing sparse modeling approaches can be cast as special examples of (1), such as sparse linear regression [30], sparse logistic regression [32], and sparse Poisson regression [26].

Given a convex regularizer, e.g., $\mathcal{R}_{\text{tgt}}(\theta) = \lambda_{\text{tgt}}||\theta||_1$ [30], we can obtain global optima in polynomial time and characterize their statistical properties. However, convex regularizers incur large estimation bias. To address this issue, several nonconvex regularizers are proposed, including the minimax concave penalty (MCP, [39]), smooth clipped absolute deviation (SCAD, [8]), and capped $\ell_1$-regularization [40]. The obtained estimator (e.g., hypothetically global optima to (1)) can achieve faster statistical rates of convergence than their convex counterparts [9, 16, 22, 34].

Despite of these superior statistical guarantees, nonconvex regularizers raise greater computational challenge than convex regularizers in high dimensions. Popular iterative algorithms for convex optimization, such as proximal gradient descent [2, 23] and coordinate descent [17, 29], no longer have strong global convergence guarantees for nonconvex optimization. Therefore, establishing statistical properties of the estimators obtained by these algorithms becomes very challenging, which explains why existing theoretical studies on computational and statistical guarantees for nonconvex regularized sparse modeling approaches are so limited until recent rise of a new area named "statistical optimization". Specifically, machine learning researchers start to incorporate certain structures of sparse modeling (e.g. restricted strong convexity, large regularization effect) into the algorithmic design and convergence analysis for optimization. This further motivates a few recent progresses: [16] propose proximal gradient algorithms for a family of nonconvex regularized estimators with a linear convergence to an approximate local optimum with suboptimal statistical guarantees; [34, 43] further propose homotopy proximal gradient and coordinate gradient descent algorithms with a linear convergence to a local optimum and optimal statistical guarantees; [9, 41] propose a multistage convex relaxation-based (also known as Difference of Convex (DC) Programming) proximal gradient algorithm, which can guarantee an approximate local optimum with optimal statistical properties. Their computational analysis further shows that within each stage of the convex relaxation, the proximal gradient algorithm achieves a (local) linear convergence to a unique sparse global optimum for the relaxed convex subproblem.

The aforementioned approaches only consider first order algorithms, such as proximal gradient descent and proximal coordinate gradient descent. The second order algorithms with theoretical guarantees are still largely missing for high dimensional nonconvex regularized sparse modeling approaches, but this does not suppress the enthusiasm of applying heuristic second order algorithms to real world problems. Some evidences have already corroborated their superior computational performance over first order algorithms (e.g. `glmnet` [10]). This further motivates our attempt towards understanding the second order algorithms in high dimensions.

In this paper, we study a multistage convex relaxation-based proximal Newton algorithm for nonconvex regularized sparse learning. This algorithm is not only highly efficient in practice, but also enjoys strong computational and statistical guarantees in theory. Specifically, by leveraging a sophisticated characterization of local restricted strong convexity and Hessian smoothness, we prove that within each stage of convex relaxation, our proposed algorithm maintains the solution sparsity, and achieves a (local) quadratic convergence, which is a significant improvement over (local) linear convergence of proximal gradient algorithm in [9] (See more details in later sections). This eventually allows us to obtain an approximate local optimum with optimal statistical properties after only a few relaxations. Numerical experiments are provided to support our theory. To the best of our knowledge, this is the first of second order based approaches for high dimensional sparse learning using convex/nonconvex regularizers with strong statistical and computational guarantees.

**Notations**: Given a vector $v \in \mathbb{R}^d$, we denote the $p$-norm as $||v||_p = (\sum_{j=1}^d |v_j|^p)^{1/p}$ for a real $p > 0$ and the number of nonzero entries as $||v||_0 = \sum_j \mathbb{1}(v_j \neq 0)$ and $v_{\backslash j} = (v_1, \ldots, v_{j-1}, v_{j+1}, \ldots, v_d)^\top \in \mathbb{R}^{d-1}$ as the subvector with the $j$-th entry removed. Given an index set $\mathcal{A} \subseteq \{1, ..., d\}$, $\mathcal{A}_\perp = \{j \mid j \in \{1, ..., d\}, j \notin \mathcal{A}\}$ is the complementary set to $\mathcal{A}$. We use $v_\mathcal{A}$ to denote a subvector of $v$ indexed by $\mathcal{A}$. Given a matrix $A \in \mathbb{R}^{d \times d}$, we use $A_{*j}$ ($A_{k*}$) to denote the $j$-th column ($k$-th row) and $\Lambda_{\max}(A)$ ($\Lambda_{\min}(A)$) as the largest (smallest) eigenvalue of $A$. We define $||A||_F^2 = \sum_j ||A_{*j}||_2^2$ and $||A||_2 = \sqrt{\Lambda_{\max}(A^\top A)}$. We denote $A_{\backslash i \backslash j}$ as the submatrix of $A$ with the $i$-th row and the $j$-th column removed, $A_{\backslash ij}$ ($A_{i \backslash j}$) as the $j$-th column ($i$-th row) of $A$ with its $i$-th ($j$-th) entry removed, and $A_{\mathcal{A}\mathcal{A}}$ as a submatrix of $A$ with both row and column indexed by $\mathcal{A}$. If $A$ is a PSD matrix, we define $||v||_A = \sqrt{v^\top A v}$ as the induced seminorm for vector $v$. We use conventional notation $\mathcal{O}(\cdot), \Omega(\cdot), \Theta(\cdot)$ to denote the limiting behavior, ignoring constant, and $\mathcal{O}_P(\cdot)$ to denote the limiting behavior in probability. $C_1, C_2, \ldots$ are denoted as generic positive constants.

## 2 DC Proximal Newton Algorithm

Throughout the rest of the paper, we assume: (1) $\mathcal{L}(\theta)$ is nonstrongly convex and twice continuously differentiable, e.g., the negative log-likelihood function of the generalized linear model (GLM); (2) $\mathcal{L}(\theta)$ takes an additive form, i.e., $\mathcal{L}(\theta) = \frac{1}{n}\sum_{i=1}^{n}\ell_i(\theta)$, where each $\ell_i(\theta)$ is associated with an observation $(x_i, y_i)$ for $i = 1, ..., n$. Take GLM as an example, we have $\ell_i(\theta) = \psi(x_i^\top \theta) - y_i x_i^\top \theta$, where $\psi$ is the cumulant function.

For nonconvex regularization, we use the capped $\ell_1$ regularizer [40] defined as

$$\mathcal{R}_{\lambda_{\text{tgt}}}(\theta) = \sum_{j=1}^{d} r_{\text{tgt}}(\theta_j) = \lambda_{\text{tgt}} \sum_{j=1}^{d} \min\{|\theta_j|, \beta\lambda_{\text{tgt}}\},$$

where $\beta > 0$ is an additional tuning parameter. Our algorithm and theory can also be extended to the SCAD and MCP regularizers in a straightforward manner [8, 39]. As shown in Figure 1, $r_{\lambda_{\text{tgt}}}(\theta_j)$ can be decomposed as the difference of two convex functions [5], i.e.,

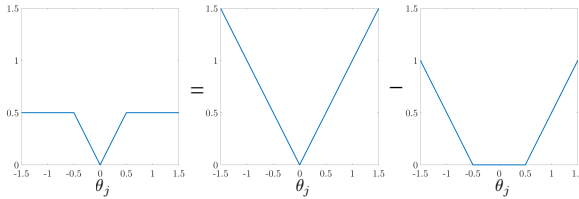

$$r_\lambda(\theta_j) = \underbrace{\lambda|\theta_j|}_{\text{convex}} - \underbrace{\max\{\lambda|\theta_j| - \beta\lambda^2, 0\}}_{\text{convex}}.$$

Figure 1: The capped $\ell_1$ regularizer is the difference of two convex functions. This allows us to relax the nonconvex regularizer based the concave duality.

This motivates us to apply the difference of convex (DC) programming approach to solve the nonconvex problem. We then introduce the DC proximal Newton algorithm, which contains three components: the multistage convex relaxation, warm initialization, and proximal Newton algorithm.

**(I) The multistage convex relaxation** is essentially a sequential optimization framework [40]. At the $(K+1)$-th stage, we have the output solution from the previous stage $\widehat{\theta}^{\{K\}}$. For notational simplicity, we define a regularization vector as $\lambda^{\{K+1\}} = (\lambda_1^{\{K+1\}}, ..., \lambda_d^{\{K+1\}})^\top$, where $\lambda_j^{\{K+1\}} = \lambda_{\text{tgt}} \cdot \mathbb{1}(|\widehat{\theta}_j^{\{K\}}| \leq \beta\lambda_{\text{tgt}})$ for all $j = 1, \ldots, d$. Let $\odot$ be the Hadamard (entrywise) product. We solve a convex relaxation of (1) at $\theta = \widehat{\theta}^{\{K\}}$ as follows,

$$\overline{\theta}^{\{K+1\}} = \underset{\theta \in \mathbb{R}^d}{\text{argmin}}\, \mathcal{F}_{\lambda^{\{K+1\}}}(\theta), \text{ where } \mathcal{F}_{\lambda^{\{K+1\}}}(\theta) = \mathcal{L}(\theta) + ||\lambda^{\{K+1\}} \odot \theta||_1, \qquad (2)$$

where $||\lambda^{\{K+1\}} \odot \theta||_1 = \sum_{j=1}^{d} \lambda_j^{\{K+1\}}|\theta_j|$. One can verify that $||\lambda^{\{K+1\}} \odot \theta||_1$ is essentially a convex relaxation of $\mathcal{R}_{\lambda_{\text{tgt}}}(\theta)$ at $\theta = \widehat{\theta}^{\{K\}}$ based on the concave duality in DC programming. We emphasis that $\overline{\theta}^{\{K\}}$ denotes the unique sparse global optimum for (2) (The uniqueness will be elaborated in later sections), and $\widehat{\theta}^{\{K\}}$ denotes the output solution for (2) when we terminate the iteration at the $K$-th convex relaxation stage. The stopping criterion will be explained later.

**(II) The warm initialization** is the first stage of DC programming, where we solve the $\ell_1$ regularized counterpart of (1),

$$\overline{\theta}^{\{1\}} = \underset{\theta \in \mathbb{R}^d}{\text{argmin}}\, \mathcal{L}(\theta) + \lambda_{\text{tgt}}||\theta||_1. \qquad (3)$$

This is an intuitive choice for sparse statistical recovery, since the $\ell_1$ regularized estimator can give us a good initialization, which is sufficiently close to $\theta^*$. Note that this is equivalent to (2) with $\lambda_j^{\{1\}} = \lambda_{\text{tgt}}$ for all $j = 1, \ldots, d$, which can be viewed as the convex relaxation of (1) at $\widehat{\theta}^{\{0\}} = \mathbf{0}$ for the first stage.

**(III) The proximal Newton algorithm** proposed in [12] is then applied to solve the convex subproblem (2) at each stage, including the warm initialization (3). For notational simplicity, we omit the stage index $\{K\}$ for all intermediate updates of $\theta$, and only use $(t)$ as the iteration index within the $K$-th stage for all $K \geq 1$. Specifically, at the $K$-th stage, given $\theta^{(t)}$ at the $t$-th iteration of the proximal Newton algorithm, we consider a quadratic approximation of (2) at $\theta^{(t)}$ as follows,

$$\mathcal{Q}(\theta; \theta^{(t)}, \lambda^{\{K\}}) = \mathcal{L}(\theta^{(t)}) + (\theta - \theta^{(t)})^\top \nabla\mathcal{L}(\theta^{(t)}) + \frac{1}{2}||\theta - \theta^{(t)}||_{\nabla^2\mathcal{L}(\theta^{(t)})}^2 + ||\lambda^{\{K\}} \odot \theta||_1, \quad (4)$$

where $||\theta - \theta^{(t)}||^2_{\nabla^2 \mathcal{L}(\theta^{(t)})} = (\theta - \theta^{(t)})^\top \nabla^2 \mathcal{L}(\theta^{(t)})(\theta - \theta^{(t)})$. We then take $\theta^{(t+\frac{1}{2})} = \arg\min_\theta \mathcal{Q}(\theta; \theta^{(t)}, \lambda^{\{K\}})$. Since $\mathcal{L}(\theta) = \frac{1}{n}\sum_{i=1}^n \ell_i(\theta)$ takes an additive form, we can avoid directly computing the $d$ by $d$ Hessian matrix in (4). Alternatively, in order to reduce the memory usage when $d$ is large, we rewrite (4) as a regularized weighted least square problem as follows

$$\mathcal{Q}(\theta; \theta^{(t)}) = \frac{1}{n}\sum_{i=1}^n w_i(z_i - x_i^\top \theta)^2 + ||\lambda^{\{K\}} \odot \theta||_1 + \text{constant}, \qquad (5)$$

where $w_i$'s and $z_i$'s are some easy to compute constants depending on $\theta^{(t)}$, $\ell_i(\theta^{(t)})$'s, $x_i$'s, and $y_i$'s.

**Remark 1.** *Existing literature has shown that* (5) *can be efficiently solved by coordinate descent algorithms in conjunction with the active set strategy [43]. See more details in [10] and Appendix B.*

For the first stage (i.e., warm initialization), we require an additional backtracking line search procedure to guarantee the descent of the objective value [12]. Specifically, we denote

$$\Delta\theta^{(t)} = \theta^{(t+\frac{1}{2})} - \theta^{(t)}.$$

Then we start from $\eta_t = 1$ and use backtracking line search to find the optimal $\eta_t \in (0, 1]$ such that the Armijo condition [1] holds. Specifically, given a constant $\mu \in (0.9, 1)$, we update $\eta_t = \mu^q$ from $q = 0$ and find the smallest integer $q$ such that

$$\mathcal{F}_{\lambda^{\{1\}}}(\theta^{(t)} + \eta_t \Delta\theta^{(t)}) \leq \mathcal{F}_{\lambda^{\{1\}}}(\theta^{(t)}) + \alpha\eta_t\gamma_t,$$

where $\alpha \in (0, \frac{1}{2})$ is a fixed constant and

$$\gamma_t = \nabla\mathcal{L}\left(\theta^{(t)}\right)^\top \cdot \Delta\theta^{(t)} + ||\lambda^{\{1\}} \odot \left(\theta^{(t)} + \Delta\theta^{(t)}\right)||_1 - ||\lambda^{\{1\}} \odot \theta^{(t)}||_1.$$

We then set $\theta^{(t+1)}$ as $\theta^{(t+1)} = \theta^{(t)} + \eta_t \Delta\theta^{(t)}$. We terminate the iterations when the following approximate KKT condition holds:

$$\omega_{\lambda^{\{1\}}}\left(\theta^{(t)}\right) := \min_{\xi \in \partial||\theta^{(t)}||_1} ||\nabla\mathcal{L}(\theta^{(t)}) + \lambda^{\{1\}} \odot \xi||_\infty \leq \varepsilon,$$

where $\varepsilon$ is a predefined precision parameter. Then we set the output solution as $\widehat{\theta}^{\{1\}} = \theta^{(t)}$. Note that $\widehat{\theta}^{\{1\}}$ is then used as the initial solution for the second stage of convex relaxation (2). The proximal Newton algorithm with backtracking line search is summarized in Algorithm 2 in Appendix.

Such a backtracking line search procedure is not necessary at $K$-th stage for all $K \geq 2$. In other words, we simply take $\eta_t = 1$ and $\theta^{(t+1)} = \theta^{(t+\frac{1}{2})}$ for all $t \geq 0$ when $K \geq 2$. This leads to more efficient updates for the proximal Newton algorithm from the second stage of convex relaxation (2). We summarize our proposed DC proximal Newton algorithm in Algorithm 1 in Appendix.

## 3 Computational and Statistical Theories

Before we present our theoretical results, we first introduce some preliminaries, including important definitions and assumptions. We define the largest and smallest $s$-sparse eigenvalues as follows.

**Definition 2.** *We define the largest and smallest $s$-**sparse eigenvalues** of $\nabla^2\mathcal{L}(\theta)$ as*

$$\rho_s^+ = \sup_{||v||_0 \leq s} \frac{v^\top \nabla^2\mathcal{L}(\theta)v}{v^\top v} \quad and \quad \rho_s^- = \inf_{||v||_0 \leq s} \frac{v^\top \nabla^2\mathcal{L}(\theta)v}{v^\top v}$$

*for any positive integer $s$. We define $\kappa_s = \frac{\rho_s^+}{\rho_s^-}$ as the $s$-sparse condition number.*

The sparse eigenvalue (SE) conditions are widely studied in high dimensional sparse modeling problems, and are closely related to restricted strong convexity/smoothness properties and restricted eigenvalue properties [22, 27, 33, 44]. For notational convenience, given a parameter $\theta \in \mathbb{R}^d$ and a real constant $R > 0$, we define a neighborhood of $\theta$ with radius $R$ as $\mathcal{B}(\theta, R) := \{\phi \in \mathbb{R}^d \mid ||\phi - \theta||_2 \leq R\}$.

Our first assumption is for the sparse eigenvalues of the Hessian matrix over a sparse domain.

**Assumption 1.** *Given $\theta \in \mathcal{B}(\theta^*, R)$ for a generic constant $R$, there exists a generic constant $C_0$ such that $\nabla^2\mathcal{L}(\theta)$ satisfies SE with parameters $0 < \rho_{s^*+2\widetilde{s}}^- < \rho_{s^*+2\widetilde{s}}^+ < +\infty$, where $\widetilde{s} \geq C_0\kappa_{s^*+2\widetilde{s}}^2 s^*$ and $\kappa_{s^*+2\widetilde{s}} = \frac{\rho_{s^*+2\widetilde{s}}^+}{\rho_{s^*+2\widetilde{s}}^-}$.*

Assumption 1 requires that $\mathcal{L}(\theta)$ has finite largest and positive smallest sparse eigenvalues, given $\theta$ is sufficiently sparse and close to $\theta^*$. Analogous conditions are widely used in high dimensional analysis [13, 14, 34, 35, 43], such as the *restricted strong convexity/smoothness* of $\mathcal{L}(\theta)$ (RSC/RSS, [6]). Given any $\theta, \theta' \in \mathbb{R}^d$, the RSC/RSS parameter can be defined as $\delta(\theta', \theta) := \mathcal{L}(\theta') - \mathcal{L}(\theta) - \nabla\mathcal{L}(\theta)^\top(\theta' - \theta)$. For notational simplicity, we define $\mathcal{S} = \{j \mid \theta_j^* \neq 0\}$ and $\mathcal{S}_\perp = \{j \mid \theta_j^* = 0\}$. The following proposition connects the SE property to the RSC/RSS property.

**Proposition 3.** *Given* $\theta, \theta' \in \mathcal{B}(\theta^*, R)$ *with* $||\theta_{\mathcal{S}_\perp}||_0 \leq \widetilde{s}$ *and* $||\theta'_{\mathcal{S}_\perp}||_0 \leq \widetilde{s}$, $\mathcal{L}(\theta)$ *satisfies*
$$\tfrac{1}{2}\rho_{s^*+2\widetilde{s}}^-\|\theta' - \theta\|_2^2 \leq \delta(\theta', \theta) \leq \tfrac{1}{2}\rho_{s^*+2\widetilde{s}}^+\|\theta' - \theta\|_2^2.$$

The proof of Proposition 3 is provided in [6], and therefore is omitted. Proposition 3 implies that $\mathcal{L}(\theta)$ is essentially strongly convex, but only over a sparse domain (See Figure 2).

The second assumption requires $\nabla^2\mathcal{L}(\theta)$ to be smooth over the sparse domain.

**Assumption 2** (Local Restricted Hessian Smoothness). *Recall that* $\widetilde{s}$ *is defined in Assumption 1. There exist generic constants* $L_{s^*+2\widetilde{s}}$ *and* $R$ *such that for any* $\theta, \theta' \in \mathcal{B}(\theta^*, R)$ *with* $||\theta_{\mathcal{S}_\perp}||_0 \leq \widetilde{s}$ *and* $||\theta'_{\mathcal{S}_\perp}||_0 \leq \widetilde{s}$, *we have* $\sup_{v\in\Omega,\ ||v||=1} v^\top(\nabla^2\mathcal{L}(\theta') - \nabla^2\mathcal{L}(\theta))v \leq L_{s^*+2\widetilde{s}}\|\theta - \theta'\|_2^2$, *where* $\Omega = \{v \mid \operatorname{supp}(v) \subseteq (\operatorname{supp}(\theta) \cup \operatorname{supp}(\theta'))\}$.

Assumption 2 guarantees that $\nabla^2\mathcal{L}(\theta)$ is Lipschitz continuous within a neighborhood of $\theta^*$ over a sparse domain. The local restricted Hessian smoothness is parallel to the local Hessian smoothness for analyzing the proximal Newton method in low dimensions [12].

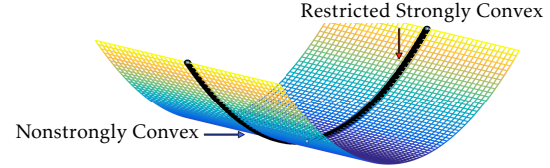

Restricted Strongly Convex

Nonstrongly Convex →

In our analysis, we set the radius $R$ as $R := \frac{\rho_{s^*+2\widetilde{s}}^-}{2L_{s^*+2\widetilde{s}}}$, where $2R = \frac{\rho_{s^*+2\widetilde{s}}^-}{L_{s^*+2\widetilde{s}}}$ is the radius of the region centered at the unique global minimizer of (2) for quadratic convergence of the proximal Newton algorithm. This is parallel to the radius in low dimensions [12], except that we restrict the parameters over the sparse domain.

Figure 2: An illustrative two dimensional example of the restricted strong convexity. $\mathcal{L}(\theta)$ is not strongly convex. But if we restrict $\theta$ to be sparse (Black Curve), $\mathcal{L}(\theta)$ behaves like a strongly convex function.

The third assumption requires the choice of $\lambda_{\text{tgt}}$ to be appropriate.

**Assumption 3.** *Given the true modeling parameter* $\theta^*$, *there exists a generic constant* $C_1$ *such that* $\lambda_{\text{tgt}} = C_1\sqrt{\frac{\log d}{n}} \geq 4||\nabla\mathcal{L}(\theta^*)||_\infty$. *Moreover, for a large enough* $n$, *we have* $\sqrt{s^*}\lambda_{\text{tgt}} \leq C_2 R\rho_{s^*+2\widetilde{s}}^-$.

Assumption 3 guarantees that the regularization is sufficiently large to eliminate irrelevant coordinates such that the obtained solution is sufficiently sparse [4, 22]. In addition, $\lambda_{\text{tgt}}$ can not be too large, which guarantees that the estimator is close enough to the true model parameter. The above assumptions are deterministic. We will verify them under GLM in the statistical analysis.

Our last assumption is on the predefined precision parameter $\varepsilon$ as follows.

**Assumption 4.** *For each stage of solving the convex relaxation subproblems* (2) *for all* $K \geq 1$, *there exists a generic constant* $C_3$ *such that* $\varepsilon$ *satisfies* $\varepsilon = \frac{C_3}{\sqrt{n}} \leq \frac{\lambda_{\text{tgt}}}{8}$.

Assumption 4 guarantees that the output solution $\widehat{\theta}^{\{K\}}$ at each stage for all $K \geq 1$ has a sufficient precision, which is critical for our convergence analysis of multistage convex relaxation.

## 3.1 Computational Theory

We first characterize the convergence for the first stage of our proposed DC proximal Newton algorithm, i.e., the warm initialization for solving (3).

**Theorem 4** (Warm Initialization, $K = 1$). *Suppose that Assumptions $1 \sim 4$ hold. After sufficiently many iterations $T < \infty$, the following results hold for all $t \geq T$:*
$$||\theta^{(t)} - \theta^*||_2 \leq R \ \text{ and } \ \mathcal{F}_{\lambda^{\{1\}}}(\theta^{(t)}) \leq \mathcal{F}_{\lambda^{\{1\}}}(\theta^*) + \frac{15\lambda_{\text{tgt}}^2 s^*}{4\rho_{s^*+2\widetilde{s}}^-},$$

*which further guarantee*

$$\eta_t = 1, \ ||\theta_{\mathcal{S}_\perp}^{(t)}||_0 \leq \widetilde{s} \ \text{ and } \ ||\theta^{(t+1)} - \overline{\theta}^{\{1\}}||_2 \leq \frac{L_{s^*+2\widetilde{s}}}{2\rho_{s^*+2\widetilde{s}}^-}||\theta^{(t)} - \overline{\theta}^{\{1\}}||_2^2,$$

*where $\overline{\theta}^{\{1\}}$ is the unique sparse global minimizer of* (3) *satisfying* $||\overline{\theta}_{\mathcal{S}_\perp}^{\{1\}}||_0 \leq \widetilde{s}$ *and* $\omega_{\lambda^{\{1\}}}(\overline{\theta}^{\{1\}}) = 0$. *Moreover, we need at most*

$$T + \log\log\left(\frac{3\rho_{s^*+2\widetilde{s}}^+}{\varepsilon}\right)$$

*iterations to terminate the proximal Newton algorithm for the warm initialization* (3)*, where the output solution $\widehat{\theta}^{\{1\}}$ satisfies*

$$||\widehat{\theta}_{\mathcal{S}_\perp}^{\{1\}}||_0 \leq \widetilde{s}, \ \omega_{\lambda^{\{1\}}}(\widehat{\theta}^{\{1\}}) \leq \varepsilon, \ \text{ and } \ ||\widehat{\theta}^{\{1\}} - \theta^*||_2 \leq \frac{18\lambda_{\text{tgt}}\sqrt{s^*}}{\rho_{s^*+2\widetilde{s}}^-}.$$

The proof of Theorem 4 is provided in Appendix C.1. Theorem 4 implies: **(I)** The objective value is sufficiently small after finite $T$ iterations of the proximal Newton algorithm, which further guarantees sparse solutions and good computational performance in all follow-up iterations. **(II)** The solution enters the ball $\mathcal{B}(\theta^*, R)$ after finite $T$ iterations. Combined with the sparsity of the solution, it further guarantees that the solution enters the region of quadratic convergence. Thus the backtracking line search stops immediately and output $\eta_t = 1$ for all $t \geq T$. **(III)** The total number of iterations is at most $\mathcal{O}(T + \log\log\frac{1}{\varepsilon})$ to achieve the approximate KKT condition $\omega_{\lambda^{\{1\}}}(\theta^{(t)}) \leq \varepsilon$, which serves as the stopping criterion of the warm initialization (3).

Given these good properties of the output solution $\widehat{\theta}^{\{1\}}$ obtained from the warm initialization, we can further show that our proposed DC proximal Newton algorithm for all follow-up stages (i.e., $K \geq 2$) achieves better computational performance than the first stage. This is characterized by the following theorem. For notational simplicity, we omit the iteration index $\{K\}$ for the intermediate updates within each stage for the multistage convex relaxation.

**Theorem 5** (Stage $K$, $K \geq 2$). *Suppose Assumptions 1 $\sim$ 4 hold. Then for all iterations $t = 1, 2, ...$ within each stage $K \geq 2$, we have*

$$||\theta_{\mathcal{S}_\perp}^{(t)}||_0 \leq \widetilde{s} \quad \text{and} \quad ||\theta^{(t)} - \theta^*||_2 \leq R,$$

*which further guarantee*

$$\eta_t = 1, \ ||\theta^{(t+1)} - \overline{\theta}^{\{K\}}||_2 \leq \frac{L_{s^*+2\widetilde{s}}}{2\rho_{s^*+2\widetilde{s}}^-}||\theta^{(t)} - \overline{\theta}^{\{K\}}||_2^2, \ \text{ and } \ \mathcal{F}_{\lambda^{\{K\}}}(\theta^{(t+1)}) < \mathcal{F}_{\lambda^{\{K\}}}(\theta^{(t)}),$$

*where $\overline{\theta}^{\{K\}}$ is the unique sparse global minimizer of* (2) *at the $K$-th stage satisfying* $||\overline{\theta}_{\mathcal{S}_\perp}^{\{K\}}||_0 \leq \widetilde{s}$ *and* $\omega_{\lambda^{\{K\}}}(\overline{\theta}^{\{K\}}) = 0$. *Moreover, we need at most*

$$\log\log\left(\frac{3\rho_{s^*+2\widetilde{s}}^+}{\varepsilon}\right).$$

*iterations to terminate the proximal Newton algorithm for the $K$-th stage of convex relaxation* (2)*, where the output solution $\widehat{\theta}^{\{K\}}$ satisfies* $||\widehat{\theta}_{\mathcal{S}_\perp}^{\{K\}}||_0 \leq \widetilde{s}$, $\omega_{\lambda^{\{K\}}}(\widehat{\theta}^{\{K\}}) \leq \varepsilon$, *and*

$$||\widehat{\theta}^{\{K\}} - \theta^*||_2 \leq C_2 \left( ||\nabla\mathcal{L}(\theta^*)_{\mathcal{S}}||_2 + \lambda_{\text{tgt}}\sqrt{\sum_{j\in\mathcal{S}}\mathbb{1}(|\theta_j^*| \leq \beta\lambda_{\text{tgt}})^2} + \varepsilon\sqrt{s^*} \right)$$
$$+ C_3 0.7^{K-1}||\widehat{\theta}^{\{1\}} - \theta^*||_2,$$

*for some generic constants $C_2$ and $C_3$.*

The proof of Theorem 5 is provided in Appendix C.2. A geometric interpretation for the computational theory of local quadratic convergence for our proposed algorithm is provided in Figure 3. From the second stage of convex relaxation (2), i.e., $K \geq 2$, Theorem 5 implies: **(I)** Within each stage, the al-

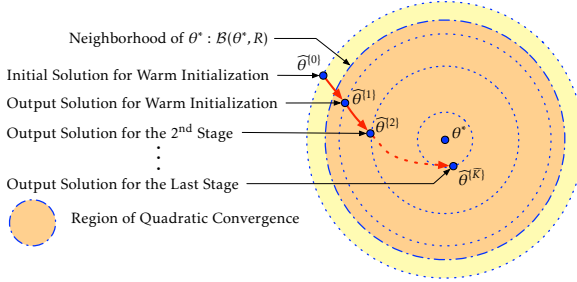

Neighborhood of $\theta^*$ : $\mathcal{B}(\theta^*, R)$

Initial Solution for Warm Initialization — $\widehat{\theta}^{[0]}$

Output Solution for Warm Initialization — $\widehat{\theta}^{[1]}$

Output Solution for the 2$^{nd}$ Stage — $\widehat{\theta}^{[2]}$   $\theta^*$

Output Solution for the Last Stage — $\widehat{\theta}^{[\overline{K}]}$

Region of Quadratic Convergence

Figure 3: A geometric interpretation of local quadratic convergence: the warm initialization enters the region of quadratic convergence (orange region) after finite iterations and the follow-up stages remain in the region of quadratic convergence. The final estimator $\widehat{\theta}^{\{\widetilde{K}\}}$ has a better estimation error than the estimator $\widehat{\theta}^{\{1\}}$ obtained from the convex warm initialization.

gorithm maintains a sparse solution throughout all iterations $t \geq 1$. The sparsity further guarantees that the SE property and the restrictive Hessian smoothness hold, which are necessary conditions for the fast convergence of the proximal Newton algorithm. **(II)** The solution is maintained in the region $\mathcal{B}(\theta^*, R)$ for all $t \geq 1$. Combined with the sparsity of the solution, we have that the solution enters the region of quadratic convergence. This guarantees that we only need to set the step size $\eta_t = 1$ and the objective value is monotonely decreasing without the sophisticated backtracking line search procedure. Thus, the proximal Newton algorithm enjoys the same fast convergence as in low dimensional optimization problems [12].

**(III)** With the quadratic convergence rate, the number of iterations is at most $\mathcal{O}(\log \log \frac{1}{\varepsilon})$ to attain the approximate KKT condition $\omega_{\lambda^{\{K\}}}(\theta^{(t)}) \leq \varepsilon$, which is the stopping criteria at each stage.

## 3.2 Statistical Theory

Recall that our computational theory relies on deterministic assumptions (Assumptions 1 $\sim$ 3). However, these assumptions involve data, which are sampled from certain statistical distribution. Therefore, we need to verify that these assumptions hold with high probability under mild data generation process of (i.e., GLM) in high dimensions in the following lemma.

**Lemma 6.** *Suppose that $x_i$'s are i.i.d. sampled from a zero-mean distribution with covariance matrix $\text{Cov}(x_i) = \Sigma$ such that $\infty > c_{\max} \geq \Lambda_{\max}(\Sigma) \geq \Lambda_{\min}(\Sigma) \geq c_{min} > 0$, and for any $v \in \mathbb{R}^d$, $v^\top x_i$ is sub-Gaussian with variance at most $a||v||_2^2$, where $c_{\max}$, $c_{\min}$, and $a$ are generic constants. Moreover, for some constant $M_\psi > 0$, at least one of the following two conditions holds: (I) The Hessian of the cumulant function $\psi$ is uniformly bounded: $||\psi''||_\infty \leq M_\psi$, or (II) The covariates are bounded $||x_i||_\infty \leq 1$, and $\mathbb{E}[\max_{|u| \leq 1}[\psi''(x^\top \theta^*) + u]^p] \leq M_\psi$ for some $p > 2$. Then Assumption 1 $\sim$ 3 hold with high probability.*

The proof of Lemma 6 is provided in Appendix F. Given that these assumptions hold with high probability, we know that the proximal Newton algorithm attains quadratic rate convergence within each stage of convex relaxation with high probability. Then we establish the statistical rate of convergence for the obtained estimator in parameter estimation.

**Theorem 7.** *Suppose the observations are generated from GLM satisfying the condition in Lemma 6 for large enough $n$ such that $n \geq C_4 s^* \log d$ and $\beta = C_5/c_{\min}$ is a constant defined in Section 2 for generic constants $C_4$ and $C_5$, then with high probability, the output solution $\widehat{\theta}^{\{K\}}$ satisfies*

$$||\widehat{\theta}^{\{K\}} - \theta^*||_2 \leq C_6 \left( \sqrt{\frac{s^*}{n}} + \sqrt{\frac{s' \log d}{n}} \right) + C_7 0.7^K \left( \sqrt{\frac{s^* \log d}{n}} \right)$$

*for generic constants $C_6$ and $C_7$, where $s' = \sum_{j \in \mathcal{S}} \mathbb{1}(|\theta_j^*| \leq \beta \lambda_{\text{tgt}})$.*

Theorem 7 is a direct result combining Theorem 5 and the analysis in [40]. As can be seen, $s'$ is essentially the number of nonzero $\theta_j$'s with smaller magnitudes than $\beta \lambda_{\text{tgt}}$, which are often considered as "weak" signals. Theorem 7 essentially implies that by exploiting the multi-stage convex relaxation framework, our proposed DC proximal Newton algorithm gradually reduces the estimation bias for "strong" signals, and eventually obtains an estimator with better statistical properties than the $\ell_1$-regularized estimator. Specifically, let $\widetilde{K}$ be the smallest integer such that after $\widetilde{K}$ stages of convex relaxation we have $C_7 0.7^{\widetilde{K}} \left( \sqrt{\frac{s^* \log d}{n}} \right) \leq C_6 \max \left\{ \sqrt{\frac{s^*}{n}}, \sqrt{\frac{s' \log d}{n}} \right\}$, which is equivalent to requiring $\widetilde{K} = \mathcal{O}(\log \log d)$. This implies the total number of the proximal Newton updates being at most $\mathcal{O}\left( T + \log \log \frac{1}{\varepsilon} \cdot (1 + \log \log d) \right)$. In addition, the obtained estimator attains the optimal

statistical properties in parameter estimation:

$$||\widehat{\theta}^{\{\widetilde{K}\}} - \theta^*||_2 \leq \mathcal{O}_P\left(\sqrt{\frac{s^*}{n}} + \sqrt{\frac{s' \log d}{n}}\right) \quad \text{v.s.} \quad ||\widehat{\theta}^{\{1\}} - \theta^*||_2 \leq \mathcal{O}_P\left(\sqrt{\frac{s^* \log d}{n}}\right). \quad (6)$$

Recall that $\widehat{\theta}^{\{1\}}$ is obtained by the warm initialization (3). As illustrated in Figure 3, this implies the statistical rate in (6) for $||\widehat{\theta}^{\{\widetilde{K}\}} - \theta^*||_2$ obtained from the multistage convex relaxation for the nonconvex regularized problem (1) is a significant improvement over $||\widehat{\theta}^{\{1\}} - \theta^*||_2$ obtained from the convex problem (3). Especially when $s'$ is small, i.e., most of nonzero $\theta_j$'s are strong signals, our result approaches the oracle bound[3] $\mathcal{O}_P\left(\sqrt{\frac{s^*}{n}}\right)$ [8] as illustrated in Figure 4.

## 4  Experiments

We compare our DC Proximal Newton (DC+PN) algorithm with two competing algorithms for solving the nonconvex regularized sparse logistic regression problem. They are accelerated proximal gradient algorithm (APG) implemented in the SPArse Modeling Software (SPAMS, coded in C++ [18]), and accelerated coordinate descent (ACD) algorithm implemented in R package gcdnet (coded in Fortran, [36]). We further optimize the active set strategy in gcdnet to boost its computational performance. To integrate these two algorithms with the multistage convex relaxation framework, we revise their source code.

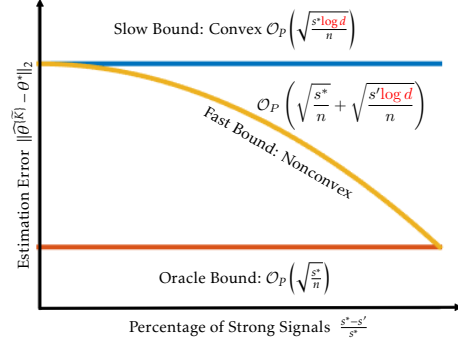

Figure 4: An illustration of the statistical rates of convergence in parameter estimation. Our obtained estimator has an error bound between the oracle bound and the slow bound from the convex problem in general. When the percentage of strong signals increases, i.e., $s'$ decreases, then our result approaches the oracle bound.

To further boost the computational efficiency at each stage of the convex relaxation, we apply the pathwise optimization [10] for all algorithms in practice. Specifically, at each stage, we use a geometrically decreasing sequence of regularization parameters $\{\lambda_{[m]} = \alpha^m \lambda_{[0]}\}_{m=1}^M$, where $\lambda_{[0]}$ is the smallest value such that the corresponding solution is zero, $\alpha \in (0,1)$ is a shrinkage parameter and $\lambda_{\text{tgt}} = \lambda_{[M]}$. For each $\lambda_{[m]}$, we apply the corresponding algorithm (DC+PN, DC+APG, and DC+ACD) to solve the nonconvex regularized problem (1). Moreover, we initialize the solution for a new regularization parameter $\lambda_{[m+1]}$ using the output solution obtained with $\lambda_{[m]}$. Such a pathwise optimization scheme has achieved tremendous success in practice [10, 15, 42]. We refer [43] for detailed discussion of pathwise optimization.

Our comparison contains 3 datasets: "madelon" ($n = 2000, d = 500$, [11]), "gisette" ($n = 2000, d = 5000$, [11]), and three simulated datasets: "sim_1k" ($d$=1000), "sim_5k" ($d$=5000), and "sim_10k" ($d$=10000) with the sample size $n = 1000$ for all three datasets. We set $\lambda_{\text{tgt}} = 0.25\sqrt{\log d/n}$ and $\beta = 0.2$ for all settings here. We generate each row of the design matrix $X$ independently from a $d$-dimensional normal distribution $\mathcal{N}(0, \Sigma)$, where $\Sigma_{jk} = 0.5^{|j-k|}$ for $j, k = 1, ..., d$. We generate $y \sim \text{Bernoulli}(1/[1 + \exp(-X\theta^*)])$, where $\theta^*$ has all 0 entries except randomly selected 20 entries. These nonzero entries are independently sampled from $\text{Uniform}(0,1)$. The stopping criteria for all algorithms are tuned such that they attain similar optimization errors.

All three algorithms are compared in wall clock time. Our DC+PN algorithm is implemented in C with double precisions and called from R by a wrapper. All experiments are performed on a computer with 2.6GHz Intel Core i7 and 16GB RAM. For each algorithm and dataset, we repeat the algorithm 10 times and report the average value and standard deviation of the wall clock time in Table 1. As can be seen, our DC+PN algorithm significantly outperforms the competing algorithms. We remark that for increasing $d$, the superiority of DC+PN over DC+ACD becomes less significant as the Newton method is more sensitive to ill conditioned problems. This can be mitigated by using a denser sequence of $\{\lambda_{[m]}\}$ along the solutions path.

We then illustrate the quadratic convergence of our DC+PN algorithm within each stage of the convex relaxation using the "sim" dataset. Specifically, we plot the gap towards the optimal objective

$\mathcal{F}_{\lambda^{\{K\}}}(\overline{\theta}^{\{K\}})$ of the $K$-th stage versus the wall clock time in Figure 5. We see that our proposed DC proximal Newton algorithm achieves quadratic convergence, which is consistent with our theory.

Table 1: Quantitive timing comparisons for the nonconvex regularized sparse logistic regression. The average values and the standard deviations (in parenthesis) of the timing performance (in seconds) over 10 random trials are presented.

|  | madelon | gisette | sim_1k | sim_5k | sim_10k |
|---|---|---|---|---|---|
| DC+PN | **1.51**($\pm 0.01$)s | **5.35**($\pm 0.11$)s | **1.07**($\pm 0.02$)s | **4.53**($\pm 0.06$)s | **8.82**($\pm 0.04$)s |
|  | obj value: 0.52 | obj value: 0.01 | obj value: 0.01 | obj value: 0.01 | obj value: 0.01 |
| DC+ACD | **5.83**($\pm 0.03$)s | **18.92**($\pm 2.25$)s | **9.46**($\pm 0.09$) s | **16.20**($\pm 0.24$) s | **19.1**($\pm 0.56$) s |
|  | obj value: 0.52 | obj value: 0.01 | obj value: 0.01 | obj value: 0.01 | obj value: 0.01 |
| DC+APG | **1.60**($\pm 0.03$)s | **207**($\pm 2.25$)s | **17.8**($\pm 1.23$) s | **111**($\pm 1.28$) s | **222**($\pm 5.79$) s |
|  | obj value: 0.52 | obj value: 0.01 | obj value: 0.01 | obj value: 0.01 | obj value: 0.01 |

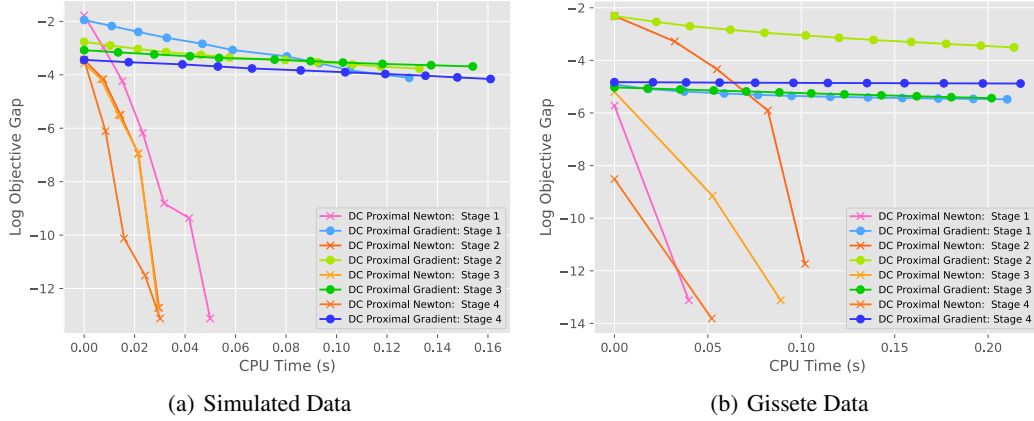

(a) Simulated Data          (b) Gissete Data

Figure 5: Timing comparisons in wall clock time. Our proposed DC proximal Newton algorithm demonstrates superior quadratic convergence and significantly outperforms the DC proximal gradient algorithm.

## 5 Discussions

We provide further discussions on the superior performance of our DC proximal Newton. There exist two major drawbacks of existing multi-stage convex relaxation based first order algorithms:

**(I)** The first order algorithms have significant computational overhead in each iteration. For example, for GLM, computing gradients requires frequently evaluating the cumulant function and its derivatives. This often involves extensive non-arithmetic operations such as $\log(\cdot)$ and $\exp(\cdot)$ functions, which naturally appear in the cumulant function and its derivates, are computationally expensive. To the best of our knowledge, even if we use some efficient numerical methods for calculating $\exp(\cdot)$ in [28, 19], the computation still need at least $10 - 30$ times more CPU cycles than basic arithmetic operations, e.g., multiplications. Our proposed DC Proximal Newton algorithm cannot avoid calculating the cumulant function and its derivatives, when computing quadratic approximations. The computation, however, is much less intense, since the convergence is quadratic.

**(II)** The first order algorithms are computationally expensive with the step size selection. Although for certain GLM, e.g., sparse logistic regression, we can choose the step size parameter as $\eta = 4\Lambda_{\max}^{-1}(\frac{1}{n}\sum_{i=1}^{n} x_i x_i^\top)$, such a step size often leads to poor empirical performance. In contrast, as our theoretical analysis and experiments suggest, the proposed DC proximal Newton algorithm needs very few line search steps, which saves much computational effort.

Some recent works on proximal Newton or inexact proximal Newton also demonstrate local quadratic convergence guarantees [37, 38]. However, the conditions there are much more stringent than the SE property in terms of the dependence on the problem dimensions. Specifically, their quadratic convergence can only be guaranteed in a much smaller neighborhood. For example, the constant nullspace strong convexity in [37], which plays the rule as the smallest sparse eigenvalue $\rho^-_{s^*+2\widetilde{s}}$ in our analysis, is as small as $\frac{1}{d}$. Note that $\rho^-_{s^*+2\widetilde{s}}$ can be (almost) independent of $d$ in our case [6]. Therefore, instead of a constant radius as in our analysis, they can only guarantee the quadratic convergence in a region with radius $\mathcal{O}\left(\frac{1}{d}\right)$, which is very small in high dimensions. A similar issue exists in [38] that the region of quadratic convergence is too small.

## Footnotes

*The work was done while the author was at Johns Hopkins University.

†The authors acknowledge support from DARPA YFA N66001-14-1-4047, NSF Grant IIS-1447639, and Doctoral Dissertation Fellowship from University of Minnesota. Correspondence to: Xingguo Li <lixx1661@umn.edu> and Tuo Zhao <tuo.zhao@isye.gatech.edu>.

[3]The oracle bound assumes that we know which variables are relevant in advance. It is not a realistic bound, but only for comparison purpose.

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
