[Supplementary Material]

# A DC Proximal Newton Algorithm

---

**Algorithm 1** DC Proximal Newton Algorithm

---

**Input:** $\widehat{\theta}^{\{0\}}, \lambda_{\text{tgt}}, \beta, \varepsilon$
**Warm Initialization:** $\widehat{\theta}^{\{1\}} \leftarrow \text{ProxNewton}(\widehat{\theta}^{\{0\}}, \lambda_{\text{tgt}}, \varepsilon), K \leftarrow 1$
**Repeat:**

$$\lambda_j^{\{K+1\}} \leftarrow \begin{cases} 0, & \text{if } |\widehat{\theta}_j^{\{K\}}| > \beta\lambda_{\text{tgt}} \\ \lambda_{\text{tgt}}, & \text{if } |\widehat{\theta}_j^{\{K\}}| \le \beta\lambda_{\text{tgt}} \end{cases}$$

$\quad t \leftarrow 0, \theta^{(0)} = \widehat{\theta}^{\{K\}}$
$\quad$**Repeat:**
$\quad\quad \theta^{(t+1)} \leftarrow \text{argmin}_\theta \, \mathcal{Q}(\theta; \theta^{(t)}, \lambda^{\{K+1\}})$
$\quad\quad t \leftarrow t+1$
$\quad$**Until** $\omega_{\lambda^{\{K+1\}}}(\theta^{(t)}) \le \varepsilon$
$\quad \widehat{\theta}^{\{K+1\}} \leftarrow \theta^{(t)}$
$\quad K \leftarrow K+1$
**Until** Convergence
**Return:** $\widehat{\theta}^{\{K\}}$.

---

---

**Algorithm 2** Proximal Newton Algorithm (ProxNewton)

---

**Input:** $\theta^{(0)}, \lambda_{\text{tgt}}, \varepsilon$
**Initialize:** $t \leftarrow 0, \lambda_j \leftarrow \lambda_{\text{tgt}}, \mu \leftarrow 0.9, \alpha \leftarrow \frac{1}{4}$
**Repeat:**
$\quad \theta^{(t+\frac{1}{2})} \leftarrow \text{argmin}_\theta \, \mathcal{Q}(\theta; \theta^{(t)}, \lambda)$
$\quad \Delta\theta^{(t)} \leftarrow \theta^{(t+\frac{1}{2})} - \theta^{(t)}$
$\quad \gamma_t \leftarrow \nabla\mathcal{L}\left(\theta^{(t)}\right)^\top \cdot \Delta\theta^{(t)} + ||\lambda \odot \left(\theta^{(t)} + \Delta\theta^{(t)}\right)||_1 - ||\lambda \odot \theta^{(t)}||_1$
$\quad \eta_t \leftarrow 1, q \leftarrow 0$
$\quad$**Repeat:**
$\quad\quad \eta_t \leftarrow \mu^q$
$\quad\quad q \leftarrow q+1$
$\quad$**Until** $\mathcal{F}_\lambda\left(\theta^{(t)} + \eta_t\Delta\theta^{(t)}\right) \le \mathcal{F}_\lambda\left(\theta^{(t)}\right) + \alpha\eta_t\gamma_t$
$\quad \theta^{(t+1)} \leftarrow \theta^{(t)} + \eta_t\Delta\theta^{(t)}$
$\quad t \leftarrow t+1$
**Until** $\omega_\lambda(\theta^{(t)}) \le \varepsilon$
**Return:** $\theta^{(t)}$.

---

# B Active Set Proximal Newton Algorithm

We first provide a brief derivation of the quadratic approximation (4) into a weighted least square problem. For notational convenience, we omit the indexes $\{K\}$ and $(t)$ for a particular iteration of a stage. Remind that we want to minimize the following $\ell_1$ regularized quadratic problem

$$\widehat{\Delta} = \underset{\Delta}{\text{argmin}} \; \widehat{\mathcal{L}}(\Delta) + ||\lambda \odot (\theta + \Delta)||_1, \tag{7}$$

where $\widehat{\mathcal{L}}(\Delta) = \Delta^\top \nabla\mathcal{L}(\theta) + \frac{1}{2}\Delta^\top\nabla^2\mathcal{L}(\theta)\Delta$. For GLM, we have

$$\mathcal{L}(\theta) = \frac{1}{2n}\sum_{i=1}^n(\psi(x_i^\top\theta) - y_ix_i^\top\theta)^2,$$

where $\psi$ is the cumulant function. Then we can rewrite the quadratic function in subproblem (7) as an *iterated reweighted least squares* [10]:

$$\widehat{\mathcal{L}}(\Delta) = \frac{1}{2n}\sum_{i=1}^n 2\left(y_i - \psi'(x_i^\top\theta)\right)x_i^\top\Delta + \psi''(x_i^\top\theta)(x_i^\top\Delta)^2 = \frac{1}{2n}\sum_{i=1}^n w_i(z_i - x_i^\top\Delta)^2 + \text{constant},$$

where $w_i = \psi''(x_i^\top \theta)$, $z_i = \frac{y_i - \psi'(x_i^\top \theta)}{\psi''(x_i^\top \theta)}$, and the constant term does not depend on $\Delta$. This indicates that (7) is equivalent a Lasso problem with reweighted least square loss function:

$$\widehat{\Delta} = \underset{\Delta}{\operatorname{argmin}} \ \frac{1}{2n} \sum_{i=1}^n w_i (z_i - x_i^\top \Delta)^2 + ||\lambda \odot (\theta + \Delta)||_1. \tag{8}$$

By solving (8), we can avoid directly computing the $d$ by $d$ Hessian matrix in (7) and significantly reduce the memory usage when $d$ is large.

We then introduce an algorithm for solving (8) leveraging the idea of active set update. The active set update scheme is very efficient in practice [10] with rigid theoretical justifications [43]. The algorithm contains two nested loops. In the *outer loop*, we separate all coordinates into two sets: active set and inactive set. Such a partition is based on some heuristic greedy scheme, such as gradient thresholding (also called strong rule [31]). Then within each iteration of the middle loop, the *inner loop* only updates coordinates in the active set in a cyclic manner until convergence, where the coordinates in the inactive set remain to be zero. After the inner loop converges, we update the active set based on a greedy selection rule that further decreases the objective value, and repeat the inner loop. Such a procedure continues until the active set no longer changes in the outer loop. We provide the algorithm description as follows and refer [43] for further details of active set based coordinate minimization. We use $(p)$ and $(l)$ to index the outer loop and inner loop respectively.

**Inner Loop**. The active set $\mathcal{A}$ and inactive set $\mathcal{A}_\perp$ are respectively set as
$$\mathcal{A} \leftarrow \{j \mid \theta_j \neq 0\} = \{j_1, j_2, \ldots, j_s\} \ \text{ and } \ \mathcal{A}_\perp \leftarrow \{j \mid j \notin \mathcal{A}\},$$
where $j_1 < j_2 < \ldots < j_s$. A coordinate-wise minimization of (8) is performed throughout the inner loop. Specifically, given $\theta^{(p,l)}$ at the $l$-th iteration of the inner loop, we solve (8) by only considering the $j$-th coordinate in the active set and fix the rest coordinates in a cyclic manner for all $j = j_1, j_2, \ldots, j_s$, i.e.,

$$\widehat{\Delta}_j = \underset{\Delta_j}{\operatorname{argmin}} \ \frac{1}{2n} \sum_{i=1}^n w_i (z_i - x_i^\top \Delta_j)^2 + |\lambda_j (\theta_j + \Delta_j)|. \tag{9}$$

Then we update $\theta_j^{(p,l+1)} = \theta_j^{(p,l)} + \widehat{\Delta}_j$. Solving (9) has a simple closed form solution by soft thresholding. Given a thresholding parameter $\tau \in (0,1)$, we terminate the inner loop when
$$\|\theta^{(p,l+1)} - \theta^{(p,l)}\|_2 \leq \tau \lambda.$$

**Outer Loop**. At the beginning of the outer loop, we initialize the active set $\mathcal{A}^{(0)}$ as follows
$$\mathcal{A}^{(0)} \leftarrow \{j \mid |\nabla_j \mathcal{L}(\theta^{(0)})| \geq (1-\nu)\lambda\} \cup \{j \mid \theta_j^{(0)} \neq 0\},$$
where $\nabla_j \mathcal{L}(\theta^{(0)})$ is the $j$-th entry of $\nabla \mathcal{L}(\theta^{(0)})$, $\nu \in (0, 0.1)$ is a thresholding parameter, and the inactive set is $\mathcal{A}_\perp^{(0)} = \{j \mid j \notin \mathcal{A}^{(0)}\}$.

Suppose at the $p$-th iteration of the outer loop, the active set is $\mathcal{A}^{(p)}$. We then perform the inner loop introduced above using $\mathcal{A}^{(p)}$ until the convergence of the inner loop and denote $\theta^{(p+1)} = \theta^{(p,l)}$, the output of the inner loop. Next, we describe how to update the active set $\mathcal{A}^{(p)}$ using the following greedy selection rule.

- We first shrink the active set as follows. The active coordinate minimization (inner loop) may yield zero solutions on $\mathcal{A}^{(p)}$. We eliminate the zero coordinates of $\theta^{(p+1)}$ from $\mathcal{A}^{(p)}$, and update the intermediate active set and inactive set respectively as
$$\mathcal{A}^{(p+\frac{1}{2})} \leftarrow \{j \in \mathcal{A}^{(p)} \mid \theta_j^{(p+1)} \neq 0\} \ \text{ and } \ \mathcal{A}_\perp^{(p+\frac{1}{2})} \leftarrow \{j \mid j \notin \mathcal{A}^{(p+\frac{1}{2})}\}.$$
- We then expand the active set as follows. Denote
$$j^{(p)} = \underset{j \in \mathcal{A}_\perp^{(p+\frac{1}{2})}}{\operatorname{argmax}} \ |\nabla_j \mathcal{L}(\theta^{(p+1)})|.$$
  The outer loop is terminated if
$$|\nabla_{j^{(p)}} \mathcal{L}(\theta^{(p+1)})| \leq (1+\delta)\lambda,$$
  where $\delta \ll 1$ is a real positive convergence parameter, e.g., $\delta = 10^{-5}$. Otherwise, we update the sets as
$$\mathcal{A}^{(p+1)} \leftarrow \mathcal{A}^{(p+\frac{1}{2})} \cup \{j^{(p)}\} \ \text{ and } \ \mathcal{A}_\perp^{(p+1)} \leftarrow \mathcal{A}_\perp^{(p+\frac{1}{2})} \backslash \{j^{(p)}\},$$

# C  Proofs of Main Results

We provide proof sketches for the main results of Theorem 4 and 5 in this section.

## C.1  Proof of Theorem 4

We provide a few important intermediate results. The first result characterizes the sparsity of the solution and an upper bound of the objective after sufficiently many iterations as follows.

**Lemma 8.** *Suppose that Assumption $1 \sim 4$ hold. After sufficiently many iterations $T < \infty$, the following results hold for all $t \geq T$:*

$$||\theta_{\mathcal{S}_\perp}^{(t)}||_0 \leq \widetilde{s} \quad and \quad \mathcal{F}_{\lambda^{\{1\}}}(\theta^{(t)}) \leq \mathcal{F}_{\lambda^{\{1\}}}(\theta^*) + \frac{15\lambda_{\mathrm{tgt}}^2 s^*}{4\rho_{s^*+2\widetilde{s}}^-}.$$

We then demonstrate the parameter estimation and quadratic convergence conditioning on the sparse solution and bounded objective error.

**Lemma 9.** *Suppose that Assumption $1 \sim 4$ hold. If $||\theta_{\mathcal{S}_\perp}^{(t)}||_0 \leq \widetilde{s}$, and $\mathcal{F}_{\lambda^{\{1\}}}(\theta^{(t)}) \leq \mathcal{F}_{\lambda^{\{1\}}}(\theta^*) + \frac{15\lambda_{\mathrm{tgt}}^2 s^*}{4\rho_{s^*+2\widetilde{s}}^-}$, we have*

$$||\theta^{(t)} - \theta^*||_2 \leq \frac{18\lambda_{\mathrm{tgt}}\sqrt{s^*}}{\rho_{s^*+2\widetilde{s}}^-} \quad and \quad ||\theta^{(t+1)} - \overline{\theta}^{\{1\}}||_2 \leq \frac{L_{s^*+2\widetilde{s}}}{2\rho_{s^*+2\widetilde{s}}^-}||\theta^{(t)} - \overline{\theta}^{\{1\}}||_2^2$$

Moreover, we characterize the sufficient number of iterations for the proximal Newton updates to achieve the approximate KKT condition.

**Lemma 10.** *Suppose that Assumption $1 \sim 4$ hold. If $||\theta_{\mathcal{S}_\perp}^{(T)}||_0 \leq \widetilde{s}$, and $\mathcal{F}_{\lambda^{\{1\}}}(\theta^{(T)}) \leq \mathcal{F}_{\lambda^{\{1\}}}(\theta^*) + \frac{15\lambda_{\mathrm{tgt}}^2 s^*}{4\rho_{s^*+2\widetilde{s}}^-}$ at some iteration $T$, we need at most*

$$T_1 \leq \log\log\left(\frac{3\rho_{s^*+2\widetilde{s}}^+}{\varepsilon}\right)$$

*extra iterations of the proximal Newton updates such that $\omega_{\lambda^{\{1\}}}(\theta^{(T+T_1)}) \leq \frac{\lambda_{\mathrm{tgt}}}{8}$.*

Combining Lemma $8 \sim 10$, we have desired results in Theorem 4.

## C.2  Proof of Theorem 5

We present a few important intermediate results that are key components of our main proof. The first result shows that in a neighborhood of the true model parameter $\theta^*$, the sparsity of the solution is preserved when we use a sparse initialization.

**Lemma 11** (Sparsity Preserving Lemma). *Suppose that Assumptions 1 and 2 hold with $\varepsilon \leq \frac{\lambda_{\mathrm{tgt}}}{8}$. Given $\theta^{(t)} \in \mathcal{B}(\theta^*, R)$ and $||\theta_{\mathcal{S}_\perp}^{(t)}||_0 \leq \widetilde{s}$, there exists a generic constant $C_1$ such that*

$$||\theta_{\mathcal{S}_\perp}^{(t+1)}||_0 \leq \widetilde{s} \quad and \quad ||\theta^{(t+1)} - \theta^{(0)}||_2 \leq \frac{C_1\lambda_{\mathrm{tgt}}\sqrt{s^*}}{\rho_{s^*+2\widetilde{s}}^-}.$$

We then show that every step of proximal Newton updates within each stage has a quadratic convergence rate to a local minimizer, if we start with a sparse solution in the refined region.

**Lemma 12.** *Suppose that Assumption $1 \sim 4$ hold. If $\theta^{(t)} \in \mathcal{B}(\theta^*, R)$ and $\left\|\theta^{(t)}_{\mathcal{S}_\perp}\right\|_0 \leq \widetilde{s}$, then for each stage $K \geq 2$, we have*

$$||\theta^{(t+1)} - \overline{\theta}^{\{K\}}||_2 \leq \frac{L_{s^*+2\widetilde{s}}}{2\rho_{s^*+2\widetilde{s}}^-}||\theta^{(t)} - \overline{\theta}^{\{K\}}||_2^2.$$

In the following, we need to use the property that the iterates $\theta^{(t)} \in \mathcal{B}(\overline{\theta}^{\{K\}}, 2R)$ instead of $\theta^{(t)} \in \mathcal{B}(\theta^*, R)$ for convergence analysis of the proximal newton method. This property holds

since we have $\theta^{(t)} \in \mathcal{B}(\theta^*, R)$ and $\overline{\theta}^{\{K\}} \in \mathcal{B}(\theta^*, R)$ simultaneously. Thus $\theta^{(t)} \in \mathcal{B}\left(\overline{\theta}^{\{K\}}, 2R\right)$, where $2R = \frac{\rho^-_{s^*+2\widetilde{s}}}{L_{s^*+2\widetilde{s}}}$ is the radius for quadratic convergence region of the proximal Newton algorithm. Next, we present a upper bound of estimation error after a proximal Newton update in terms of the estimation error before the update.

**Lemma 13.** *Suppose that Assumption 1 $\sim$ 4 hold. If $||\theta^{(t)}_{\mathcal{S}_\perp}||_0 \leq \widetilde{s}$ and $\theta^{(t)} \in \mathcal{B}(\overline{\theta}^{\{K\}}, 2R)$, then for each stage $K \geq 2$, we have*

$$||\theta^{(t+1)} - \theta^{(t)}||_2 \leq \frac{3}{2}||\theta^{(t)} - \overline{\theta}^{\{K\}}||_2.$$

The following lemma demonstrates that the step size parameter is simply 1 if the the sparse solution is in the refined region.

**Lemma 14.** *Suppose that Assumption 1 $\sim$ 4 hold. If $\theta^{(t)} \in \mathcal{B}(\overline{\theta}^{\{K\}}, 2R)$ and $||\theta^{(t)}_{\mathcal{S}_\perp}||_0 \leq \widetilde{s}$ at each stage $K \geq 2$ with $\frac{1}{4} \leq \alpha < \frac{1}{2}$, then $\eta_t = 1$. Further, we have*

$$\mathcal{F}_{\lambda^{\{K\}}}(\theta^{(t+1)}) \leq \mathcal{F}_{\lambda^{\{K\}}}(\theta^{(t)}) + \frac{1}{4}\gamma_t.$$

Moreover, we present a critical property of $\gamma_t$.

**Lemma 15.** *Denote $\Delta\theta^{(t)} = \theta^{(t)} - \theta^{(t+1)}$ and*

$$\gamma_t = \nabla\mathcal{L}\left(\theta^{(t)}\right)^\top \cdot \Delta\theta^{(t)} + \mathcal{R}_{\lambda^{\{K\}}}\left(\theta^{(t)} + \Delta\theta^{(t)}\right) - \mathcal{R}_{\lambda^{\{K\}}}\left(\theta^{(t)}\right).$$

*Then we have $\gamma_t \leq -||\Delta\theta^{(t)}||^2_{\nabla^2\mathcal{L}(\theta^{(t)})}$.*

In addition, we present the sufficient number of iterations for each convex relaxation stage to achieve the approximate KKT condition.

**Lemma 16.** *Suppose that Assumption 1 $\sim$ 4 hold. To achieve the approximate KKT condition $\omega_{\lambda^{\{K\}}}\left(\theta^{(t)}\right) \leq \varepsilon$ for any $\varepsilon > 0$ at each stage $K \geq 2$, the number of iteration for proximal Newton updates is at most*

$$\log\log\left(\frac{3\rho^+_{s^*+2\widetilde{s}}}{\varepsilon}\right).$$

We further present the contraction of the estimation error along consecutive stages, which is a direct result from oracle statistical rate in [9].

**Lemma 17.** *Suppose that Assumption 1 $\sim$ 4 hold. Then there exists a generic constant $c_1$ such that the output solution for all $K \geq 2$ satisfy*

$$||\widehat{\theta}^{\{K\}} - \theta^*||_2 \leq c_1\left(||\nabla\mathcal{L}(\theta^*)_{\mathcal{S}}||_2 + \lambda_{\text{tgt}}\sqrt{\sum_{j\in\mathcal{S}}\mathbb{1}(|\theta^*_j| \leq \beta)^2} + \varepsilon\sqrt{s^*}\right) + 0.7||\widehat{\theta}^{\{K-1\}} - \theta^*||_2.$$

Combining Lemma 11 – Lemma 15, we have the quadratic convergence of the proximal Newton algorithm within each convex relaxation stage. The rest of the results hold by further combining Lemma 16 and recursively applying Lemma 17.

# D    Proof of Intermediate Results for Theorem 5

We first introduce an important notion that is closely related with the SE property is defined as follows.

**Definition 18.** *We denote the local $\ell_1$ cone as*

$$\mathcal{C}(s, \vartheta, R) := \left\{v, \theta : \mathcal{S} \subseteq \mathcal{M}, |\mathcal{M}| \leq s, ||v_{\mathcal{M}_\perp}||_1 \leq \vartheta||v_{\mathcal{M}}||_1, ||\theta - \theta^*||_2 \leq R\right\}.$$

*Then we define the largest and smallest **localized restricted eigenvalues** (LRE) as*

$$\tau^+_{s,\vartheta,R} = \sup_{u,\theta}\left\{\frac{v^\top\nabla^2\mathcal{L}(\theta)v}{v^\top v} : (v, \theta) \in \mathcal{C}(s, \vartheta, R)\right\},$$

$$\tau^-_{s,\vartheta,R} = \inf_{u,\theta}\left\{\frac{v^\top\nabla^2\mathcal{L}(\theta)v}{v^\top v} : (v, \theta) \in \mathcal{C}(s, \vartheta, R)\right\}.$$

The following proposition demonstrate the relationships between SE and LRE. The proof can be found in [6].

**Proposition 19.** *Given any* $\theta, \theta' \in \mathcal{C}(s, \vartheta, R) \cap \mathcal{B}(\theta^*, R)$, *we have*

$$\phi_1 \tau_{s,\vartheta,R}^- \leq \rho_s^- \leq \phi_2 \tau_{s,\vartheta,R}^-, \quad and \quad \psi_1 \tau_{s,\vartheta,R}^+ \leq \rho_s^+ \leq \psi_2 \tau_{s,\vartheta,R}^+.$$

*where* $\phi_1$, $\phi_2$, $\psi_1$, *and* $\psi_2$ *are constants.*

### D.1 Proof of Lemma 11

We first demonstrate the sparsity of the update. For notational convenience, we omit the index $\{K\}$ here. Since $\theta^{(t+1)}$ is the minimizer to the proximal newton problem, we have

$$\nabla^2 \mathcal{L}(\theta^{(t)})(\theta^{(t+1)} - \theta^{(t)}) + \nabla \mathcal{L}(\theta^{(t)}) + \lambda \odot \xi^{(t+1)} = 0,$$

where $\xi^{(t+1)} \in \partial \|\theta^{(t+1)}\|_1$.

It follows from [9] that if Assumptions 3 holds, then we have $\min_{j \in \mathcal{S}'_\perp} \{\lambda_j\} \geq \lambda_{\text{tgt}}/2$ for some set $\mathcal{S}' \supset \mathcal{S}$ with $|\mathcal{S}'| \leq 2s^*$. Then the analysis of sparsity of can be performed through $\lambda_{\text{tgt}}$ directly.

We then consider the following decomposition

$$\nabla^2 \mathcal{L}(\theta^{(t)})(\theta^{(t+1)} - \theta^{(t)}) + \nabla \mathcal{L}(\theta^{(t)})$$
$$= \underbrace{\nabla^2 \mathcal{L}(\theta^{(t)})(\theta^{(t+1)} - \theta^*)}_{V_1} + \underbrace{\nabla^2 \mathcal{L}(\theta^{(t)})(\theta^* - \theta^{(t)})}_{V_2} + \underbrace{\nabla \mathcal{L}(\theta^{(t)}) - \nabla \mathcal{L}(\theta^*)}_{V_3} + \underbrace{\nabla \mathcal{L}(\theta^*)}_{V_4}.$$

We then consider the following sets:

$$A_i = \{j \in \mathcal{S}'_\perp \ : \ |(V_i)_j| \geq \lambda_{\text{tgt}}/4\}, \quad \text{for all } i \in \{1, 2, 3, 4\}.$$

**Set $\mathcal{A}_2$.** We have $\mathcal{A}_2 = \left\{ j \in \mathcal{S}'_\perp : |(\nabla^2 \mathcal{L}(\theta^{(t)})(\theta^* - \theta^{(t)}))_j| \geq \lambda_{\text{tgt}}/4 \right\}$. Consider a subset $\mathcal{S}' \subset \mathcal{A}_2$ with $|\mathcal{S}'| = s' \leq \widetilde{s}$. Suppose we choose a vector $v \in \mathbb{R}^d$ such that $\|v\|_\infty = 1$ and $\|v\|_0 = s'$ with $s' \lambda_{\text{tgt}}/4 \leq v^\top \nabla^2 \mathcal{L}(\theta^{(t)})(\theta^* - \theta^{(t)})$. Then we have

$$s' \lambda_{\text{tgt}}/4 \leq v^\top \nabla^2 \mathcal{L}(\theta^{(t)})(\theta^* - \theta^{(t)}) \leq \|v(\nabla^2 \mathcal{L}(\theta^{(t)}))^{\frac{1}{2}}\|_2 \|(\nabla^2 \mathcal{L}(\theta^{(t)}))^{\frac{1}{2}}(\theta^* - \theta^{(t)})\|_2$$

$$\overset{(i)}{\leq} \sqrt{\rho_{s^*+2\widetilde{s}}^+ \rho_{s'}^+} \|v\|_2 \|\theta^* - \theta^{(t)}\|_2 \overset{(ii)}{\leq} \sqrt{s' \rho_{s^*+2\widetilde{s}}^+ \rho_{s'}^+} \|\theta^* - \theta^{(t)}\|_2$$

$$\overset{(iii)}{\leq} \frac{C' \sqrt{s' \rho_{s^*+2\widetilde{s}}^+ \rho_{s'}^+} \lambda_{\text{tgt}} \sqrt{s^*}}{\rho_{s^*+2\widetilde{s}}^-}, \tag{10}$$

where $(i)$ is from the SE properties, $(ii)$ is from the definition of $v$, and $(iii)$ is from $\|\theta^{(t)} - \theta^*\|_2 \leq C' \lambda_{\text{tgt}} \sqrt{s^*}/\rho_{s^*+2\widetilde{s}}^-$. Then (10) implies

$$s' \leq \frac{C_2 \rho_{s^*+2\widetilde{s}}^+ \rho_{s'}^+ s^*}{(\rho_{s^*+2\widetilde{s}}^-)^2} \leq C_2 \kappa_{s^*+2\widetilde{s}}^2 s^*, \tag{11}$$

where the last inequality is from the fact that $s' = |\mathcal{S}'|$ achieves the maximum possible value such that $s' \leq \widetilde{s}$ for any subset $\mathcal{S}'$ of $\mathcal{A}_2$. (11) implies that $s' < \widetilde{s}$, so wo must have $\mathcal{S}' = \mathcal{A}_2$ to attain the maximum. Then we have

$$|\mathcal{A}_2| = s' \leq C_2 \kappa_{s^*+2\widetilde{s}}^2 s^*.$$

**Set $\mathcal{A}_3$.** We have $\mathcal{A}_3 = \left\{ j \in \mathcal{S}'_\perp : \left| (\nabla \mathcal{L}(\theta^{(t)}) - \nabla \mathcal{L}(\theta^*))_i \right| \geq \lambda_{\text{tgt}}/4 \right\}$. Suppose we choose a vector $v \in \mathbb{R}^d$ such that $\|v\|_\infty = 1$, $\|v\|_0 = |\mathcal{A}_3|$ and

$$v^\top \left( \nabla \mathcal{L}(\theta^{(t)}) - \nabla \mathcal{L}(\theta^*) \right) = \sum_{i \in \mathcal{A}_3} v_i \left( \nabla \mathcal{L}(\theta^{(t)}) - \nabla \mathcal{L}(\theta^*) \right)_i = \sum_{i \in \mathcal{A}_3} \left| \left( \nabla \mathcal{L}(\theta^{(t)}) - \nabla \mathcal{L}(\theta^*) \right)_i \right|$$

$$\geq \lambda_{\text{tgt}} |\mathcal{A}_3|/4. \tag{12}$$

Then we have

$$v^\top \left( \nabla \mathcal{L}(\theta^{(t)}) - \nabla \mathcal{L}(\theta^*) \right) \leq \|v\|_2 \|\nabla \mathcal{L}(\theta^{(t)}) - \nabla \mathcal{L}(\theta^*)\|_2 \overset{(i)}{\leq} \sqrt{|\mathcal{A}_3|} \cdot \|\nabla \mathcal{L}(\theta^{(t)}) - \nabla \mathcal{L}(\theta^*)\|_2$$

$$\overset{(ii)}{\leq} \rho_{s^*+2\widetilde{s}}^+ \sqrt{|\mathcal{A}_3|} \cdot \|\theta^{(t)} - \theta^*\|_2, \tag{13}$$

where $(i)$ is from the definition of $v$, and $(ii)$ is from the mean value theorem and analogous argument for $\mathcal{A}_2$.

Combining (12) and (13), we have

$$\lambda_{\text{tgt}}|\mathcal{A}_3| \leq 4\rho^+_{s^*+2\widetilde{s}}\sqrt{|\mathcal{A}_3|} \cdot ||\theta - \theta^*||_2 \overset{(i)}{\leq} 8\lambda_{\text{tgt}}\kappa_{s^*+2\widetilde{s}}\sqrt{3s^*|\mathcal{A}_3|}$$

where $(i)$ is from $||\theta^{(t)} - \theta^*||_2 \leq C'\lambda_{\text{tgt}}\sqrt{s^*}/\rho^-_{s^*+2\widetilde{s}}$ and definition of $\kappa_{s^*+2\widetilde{s}} = \rho^+_{s^*+2\widetilde{s}}/\rho^-_{s^*+2\widetilde{s}}$. This implies

$$|\mathcal{A}_3| \leq C_3\kappa^2_{s^*+2\widetilde{s}}s^*.$$

**Set $A_4$.** By Assumption 3 and $\lambda_{\text{tgt}} \geq 4||\nabla\mathcal{L}(\theta^*)||_\infty$, we have

$$0 \leq |V_4| \leq \sum_{i \in \mathcal{S}^*_\perp} \frac{4}{\lambda_{\text{tgt}}}|(\nabla\mathcal{L}(\theta^*))_i| \cdot \mathbb{1}(|(\nabla\mathcal{L}(\theta^*))_i| > \lambda_{\text{tgt}}/(4)) = \sum_{i \in \mathcal{S}^*_\perp} \frac{4}{\lambda_{\text{tgt}}}|(\nabla\mathcal{L}(\theta^*))_i| \cdot 0 = 0,$$

(14)

**Set $A_1$.** From Lemma 20, we have $\mathcal{F}_\lambda(\theta^{(t+1)}) \leq \mathcal{F}_\lambda(\theta^*) + \frac{\lambda_{\text{tgt}}}{4}||\theta^{(t+1)} - \theta^*||_1$. This implies

$$\mathcal{L}(\theta^{(t+1)}) - \mathcal{L}(\theta^*) \leq \lambda_{\text{tgt}}(||\theta^*||_1 - ||\theta^{(t+1)}||_1) + \frac{\lambda_{\text{tgt}}}{4}||\theta^{(t+1)} - \theta^*||_1$$

$$= \lambda_{\text{tgt}}(||\theta^*_{\mathcal{S}'}||_1 - ||\theta^{(t+1)}_{\mathcal{S}'}||_1 - ||\theta^{(t+1)}_{\mathcal{S}'_\perp}||_1) + \frac{\lambda_{\text{tgt}}}{4}||\theta^{(t+1)} - \theta^*||_1$$

$$\leq \frac{5\lambda_{\text{tgt}}}{4}||\theta^{(t+1)}_{\mathcal{S}'} - \theta^*_{\mathcal{S}'}||_1 - \frac{3\lambda_{\text{tgt}}}{4}||\theta^{(t+1)}_{\mathcal{S}'_\perp} - \theta^*_{\mathcal{S}'_\perp}||_1. \quad (15)$$

where the equality holds since $\theta^*_{\mathcal{S}'_\perp} = 0$. On the other hand, we have

$$\mathcal{L}(\theta^{(t+1)}) - \mathcal{L}(\theta^*) \overset{(i)}{\geq} \nabla\mathcal{L}(\theta^*)(\theta^{(t+1)} - \theta^*) \geq -||cL(\theta^*)||_\infty||\theta^{(t+1)} - \theta^*||_1$$

$$\overset{(ii)}{\geq} -\frac{\lambda_{\text{tgt}}}{4}||\theta^{(t+1)} - \theta^*||_1 = -\frac{\lambda_{\text{tgt}}}{4}||\theta^{(t+1)}_{\mathcal{S}'} - \theta^*_{\mathcal{S}'}||_1 - \frac{\lambda_{\text{tgt}}}{4}||\theta^{(t+1)}_{\mathcal{S}'_\perp} - \theta^*_{\mathcal{S}'_\perp}||_1, \quad (16)$$

where $(i)$ is from the convexity of $\mathcal{L}$ and $(ii)$ is from Assumption 3. Combining (15) and (16), we have

$$||\theta^{(t+1)}_{\mathcal{S}'_\perp} - \theta^*_{\mathcal{S}'_\perp}||_1 \leq 3||\theta^{(t+1)}_{\mathcal{S}'} - \theta^*_{\mathcal{S}'}||_1,$$

which implies that $\theta^{(t+1)} - \theta^* \in \mathcal{C}(s^*, 3, R)$ with respect to the set $\mathcal{S}'$.

We have $\mathcal{A}_4 = \{j \in \mathcal{S}'_\perp : |(\nabla^2\mathcal{L}(\theta^{(t)})(\theta^* - \theta^{(t+1)}))_j| \geq \lambda_{\text{tgt}}/4\}$. Consider a subset $\mathcal{S}' \subset \mathcal{A}_2$ with $|\mathcal{S}'| = s' \leq \widetilde{s}$ and a vector $v \in \mathbb{R}^d$ similar to that in $\mathcal{A}_2$. Then we have

$$s'\lambda_{\text{tgt}}/4 \leq v^\top\nabla^2\mathcal{L}(\theta^{(t)})(\theta^{(t+1)} - \theta^*) \leq ||v(\nabla^2\mathcal{L}(\theta^{(t)}))^{\frac{1}{2}}||_2||(\nabla^2\mathcal{L}(\theta^{(t)}))^{\frac{1}{2}}(\theta^{(t+1)} - \theta^*)||_2$$

$$\overset{(i)}{\leq} c_1\sqrt{\rho^+_{s^*+2\widetilde{s}}\rho^+_{s'}}||v||_2||\theta^* - \theta^{(t+1)}||_2 \overset{(ii)}{\leq} c_1\sqrt{s'\rho^+_{s^*+2\widetilde{s}}\rho^+_{s'}}||\theta^* - \theta^{(t+1)}||_2$$

$$\overset{(iii)}{\leq} \frac{c_2\sqrt{s'\rho^+_{s^*+2\widetilde{s}}\rho^+_{s'}}\lambda_{\text{tgt}}\sqrt{s^*}}{\rho^-_{s^*+2\widetilde{s}}},, \quad (17)$$

where $(i)$ is from SE condition and Proposition 19, $(ii)$ is from the definition of $v$, and $(iii)$ is from $||\theta^{(t+1)} - \theta^*||_2 \leq C'\lambda_{\text{tgt}}\sqrt{s^*}/\rho^-_{s^*+2\widetilde{s}}$. Following analogous argument in for $A_2$, we have

$$|A_1| \leq C_1\kappa^2_{s^*+2\widetilde{s}}s^*.$$

Combining the results for Set $A_1 \sim A_4$, we have that there exists some constant $C_0$ such that

$$||\theta^{(t+\frac{1}{2})}_{\mathcal{S}_\perp}||_0 \leq C_0\kappa^2_{s^*+2\widetilde{s}}s^* \leq \widetilde{s}.$$

From Lemma 14, we further have that the step size satisfies $\eta_t = 1$, then we have $\theta^{(t+1)} = \theta^{(t+\frac{1}{2})}$. The estimation error follows directly from Lemma 21.

## D.2 Proof of Lemma 12

For notational simplicity, we introduce the following proximal operator,

$$\text{prox}_r^{H,g}(\theta) = \text{argmin}_{\theta'} r(\theta') + g^\top(\theta' - \theta) + \frac{1}{2}||\theta' - \theta||_H^2.$$

Then we have

$$\theta^{(t+1)} = \text{prox}_{\mathcal{R}_{\lambda^{\{K\}}}^{\ell_1}(\theta^{(t)})}^{\nabla^2 \mathcal{L}(\theta^{(t)}), \nabla \mathcal{L}(\theta^{(t)})} \left(\theta^{(t)}\right).$$

By Lemma 11, we have

$$||\theta_{\mathcal{S}_\perp}^{(t+1)}||_0 \leq \widetilde{s}.$$

By the KKT condition of function $\min \mathcal{F}_{\lambda^{\{K\}}}$, i.e., $-\nabla \mathcal{L}(\overline{\theta}^{\{K\}}) \in \partial \mathcal{R}_{\lambda^{\{K\}}}^{\ell_1}(\overline{\theta}^{\{K\}})$, we also have

$$\overline{\theta}^{\{K\}} = \text{prox}_{\mathcal{R}_{\lambda^{\{K\}}}^{\ell_1}(\overline{\theta}^{\{K\}})}^{\nabla^2 \mathcal{L}(\theta^{(t)}), \nabla \mathcal{L}(\overline{\theta}^{\{K\}})} \left(\overline{\theta}^{\{K\}}\right).$$

By monotonicity of sub-gradient of a convex function, we have the *strictly non-expansive* property: for any $\theta, \theta' \in \mathbb{R}$, let $u = \text{prox}_r^{H,g}(\theta)$ and $v = \text{prox}_r^{H,g'}(\theta')$, then

$$(u-v)^\top H(\theta - \theta') - (u-v)^\top(g - g') \geq \|u - v\|_H^2.$$

Thus by the strictly non-expansive property of the proximal operator, we obtain

$$||\theta^{(t+1)} - \overline{\theta}^{\{K\}}||_{\nabla^2 \mathcal{L}(\overline{\theta}^{\{K\}})}^2$$

$$\leq \left(\theta^{(t+1)} - \overline{\theta}^{\{K\}}\right)^\top \left[\nabla^2 \mathcal{L}(\theta^{(t)})\left(\theta^{(t)} - \overline{\theta}^{\{K\}}\right) + \left(\nabla \mathcal{L}(\overline{\theta}^{\{K\}}) - \nabla \mathcal{L}(\theta^{(t)})\right)\right]$$

$$\leq ||\theta^{(t+1)} - \overline{\theta}^{\{K\}}||_2 \left\|\nabla^2 \mathcal{L}(\theta^{(t)})\left(\theta^{(t)} - \overline{\theta}^{\{K\}}\right) + \left(\nabla \mathcal{L}(\overline{\theta}^{\{K\}}) - \nabla \mathcal{L}(\theta^{(t)})\right)\right\|_2. \quad (18)$$

Note that both $||\theta^{(t+1)}||_0 \leq \widetilde{s}$ and $||\overline{\theta}^{\{K\}}||_0 \leq \widetilde{s}$. On the other hand, from the SE properties, we have

$$||\theta^{(t+1)} - \overline{\theta}^{\{K\}}||_{\nabla^2 \mathcal{L}(\overline{\theta}^{\{K\}})}^2 = (\theta^{(t+1)} - \overline{\theta}^{\{K\}})^\top \nabla^2 \mathcal{L}(\overline{\theta}^{\{K\}})(\theta^{(t+1)} - \overline{\theta}^{\{K\}})$$

$$\geq \rho_{s^*+2\widetilde{s}}^- ||\theta^{(t+1)} - \overline{\theta}^{\{K\}}||_2^2. \quad (19)$$

Combining (18) and (19), we have

$$\left\|\theta^{(t+1)} - \overline{\theta}^{\{K\}}\right\|_2$$

$$\leq \frac{1}{\rho_{s^*+2\widetilde{s}}^-} \left\|\nabla^2 \mathcal{L}(\theta^{(t)})\left(\theta^{(t)} - \overline{\theta}^{\{K\}}\right) + \left(\nabla \mathcal{L}(\overline{\theta}^{\{K\}}) - \nabla \mathcal{L}(\theta^{(t)})\right)\right\|_2$$

$$= \frac{1}{\rho_{s^*+2\widetilde{s}}^-} \left\|\int_0^1 \left[\nabla^2 \mathcal{L}\left(\theta^{(t)} + \tau\left(\overline{\theta}^{\{K\}} - \theta^{(t)}\right)\right) - \nabla^2 \mathcal{L}\left(\theta^{(t)}\right)\right] \cdot \left(\overline{\theta}^{\{K\}} - \theta^{(t)}\right) d\tau\right\|_2$$

$$\leq \frac{1}{\rho_{s^*+2\widetilde{s}}^-} \int_0^1 \left\|\left[\nabla^2 \mathcal{L}\left(\theta^{(t)} + \tau\left(\overline{\theta}^{\{K\}} - \theta^{(t)}\right)\right) - \nabla^2 \mathcal{L}\left(\theta^{(t)}\right)\right] \cdot \left(\overline{\theta}^{\{K\}} - \theta^{(t)}\right)\right\|_2 d\tau$$

$$\leq \frac{L_{s^*+2\widetilde{s}}}{2\rho_{s^*+2\widetilde{s}}^-} \left\|\theta^{(t)} - \overline{\theta}^{\{K\}}\right\|_2^2,$$

where the last inequality is from the local restricted Hessian smoothness of $\mathcal{L}$. Then we finish the proof by the definition of $R$.

## D.3 Proof of Lemma 14

Suppose the step size $\eta_t < 1$. Note that we do not need the step size to be $\eta_t = 1$ in Lemma 11 and Lemma 12. We denote $\Delta\theta^{(t)} = \theta^{(t+\frac{1}{2})} - \theta^{(t)}$. Then we have

$$\left\|\Delta\theta^{(t)}\right\|_2 \overset{(ii)}{\leq} \left\|\theta^{(t)} - \overline{\theta}^{\{K\}}\right\|_2 + \left\|\theta^{(t+\frac{1}{2})} - \overline{\theta}^{\{K\}}\right\|_2 \overset{(ii)}{\leq} \left\|\theta^{(t)} - \overline{\theta}^{\{K\}}\right\|_2 + \frac{L_{s^*+2\widetilde{s}}}{2\rho_{s^*+2\widetilde{s}}^-} \left\|\theta^{(t)} - \overline{\theta}^{\{K\}}\right\|_2^2$$

$$\overset{(iii)}{\leq} \frac{3}{2} \left\|\theta^{(t)} - \overline{\theta}^{\{K\}}\right\|_2, \quad (20)$$

where $(i)$ is from triangle inequality, $(ii)$ is from Lemma 12, and $(iii)$ is from $\left\|\theta^{(t)} - \overline{\theta}^{\{K\}}\right\|_2 \leq R \leq \frac{\rho^-_{s^*+2\widetilde{s}}}{L_{s^*+2\widetilde{s}}}$.

By Lemma 11, we have

$$\left\|\Delta\theta^{(t)}_{\mathcal{S}_\perp}\right\|_0 \leq 2\widetilde{s}.$$

To show $\eta_t = 1$, it is now suffice to demonstrate that

$$\mathcal{F}_{\lambda^{\{K\}}}(\theta^{(t+\frac{1}{2})}) - \mathcal{F}_{\lambda^{\{K\}}}(\theta^{(t)}) \leq \frac{1}{4}\gamma_t.$$

By expanding $\mathcal{F}_{\lambda^{\{K\}}}$, we have

$$\mathcal{F}_{\lambda^{\{K\}}}(\theta^{(t)} + \Delta\theta^{(t)}) - \mathcal{F}_{\lambda^{\{K\}}}(\theta^{(t)})$$
$$= \mathcal{L}(\theta^{(t)} + \Delta\theta^{(t)}) - \mathcal{L}(\theta^{(t)}) + \mathcal{R}^{\ell_1}_{\lambda^{\{K\}}}(\theta^{(t)} + \Delta\theta^{(t)}) - \mathcal{R}^{\ell_1}_{\lambda^{\{K\}}}(\theta^{(t)})$$
$$\overset{(i)}{\leq} \nabla\mathcal{L}(\theta^{(t)})^\top \Delta\theta^{(t)} + \frac{1}{2}\Delta(\theta^{(t)})^\top \nabla^2\mathcal{L}(\theta)\Delta\theta^{(t)} + \frac{L_{s^*+2\widetilde{s}}}{6}\left\|\Delta\theta^{(t)}\right\|_2^3$$
$$\qquad\qquad\qquad\qquad\qquad + \mathcal{R}^{\ell_1}_{\lambda^{\{K\}}}(\theta^{(t)} + \Delta\theta^{(t)}) - \mathcal{R}^{\ell_1}_{\lambda^{\{K\}}}(\theta^{(t)})$$
$$\overset{(ii)}{\leq} \gamma_t - \frac{1}{2}\gamma_t + \frac{L_{s^*+2\widetilde{s}}}{6}\left\|\Delta\theta^{(t)}\right\|_2^3 \overset{(iii)}{\leq} \frac{1}{2}\gamma_t + \frac{L_{s^*+2\widetilde{s}}}{6\rho^-_{s^*+2\widetilde{s}}}\left\|\Delta\theta^{(t)}\right\|_{\nabla^2\mathcal{L}(\theta)}\left\|\Delta\theta^{(t)}\right\|_2$$
$$\overset{(iv)}{\leq} \left(\frac{1}{2} - \frac{L_{s^*+2\widetilde{s}}}{6\rho^-_{s^*+2\widetilde{s}}}\left\|\Delta\theta^{(t)}\right\|_2\right)\gamma_t \overset{(v)}{\leq} \frac{1}{4}\gamma_t,$$

where $(i)$ is from the restricted Hessian smooth condition, $(ii)$ and $(iv)$ are from Lemma 15, $(iii)$ is from the same argument of (19), and $(v)$ is from (20), $\gamma_t < 0$, and $\left\|\theta^{(t)} - \overline{\theta}^{\{K\}}\right\|_2 \leq R \leq \frac{\rho^-_{s^*+2\widetilde{s}}}{L_{s^*+2\widetilde{s}}}$. This implies $\theta^{(t+1)} = \theta^{(t+\frac{1}{2})}$.

### D.4 Proof of Lemma 15

We denote $H = \nabla^2\mathcal{L}(\theta^{(t)})$. Since $\Delta\theta^{(t)}$ is the solution for

$$\min_{\Delta\theta^{(t)}} \nabla\mathcal{L}\left(\theta^{(t)}\right)^\top \cdot \Delta\theta^{(t)} + \frac{1}{2}\left\|\Delta\theta^{(t)}\right\|_H^2 + \mathcal{R}^{\ell_1}_{\lambda^{\{K\}}}\left(\theta^{(t)} + \Delta\theta^{(t)}\right)$$

then for any $\eta_t \in (0, 1]$, we have

$$\eta_t\nabla\mathcal{L}\left(\theta^{(t)}\right)^\top \cdot \Delta\theta^{(t)} + \frac{\eta_t^2}{2}\left\|\Delta\theta^{(t)}\right\|_H^2 + \mathcal{R}^{\ell_1}_{\lambda^{\{K\}}}\left(\theta^{(t)} + \eta_t\Delta\theta^{(t)}\right)$$
$$\geq \nabla\mathcal{L}\left(\theta^{(t)}\right)^\top \cdot \Delta\theta^{(t)} + \frac{1}{2}\left\|\Delta\theta^{(t)}\right\|_H^2 + \mathcal{R}^{\ell_1}_{\lambda^{\{K\}}}\left(\theta^{(t)} + \Delta\theta^{(t)}\right)$$

By the convexity of $\mathcal{R}^{\ell_1}_{\lambda^{\{K\}}}$, we have

$$\eta_t\nabla\mathcal{L}\left(\theta^{(t)}\right)^\top \cdot \Delta\theta^{(t)} + \frac{\eta_t^2}{2}\left\|\Delta\theta^{(t)}\right\|_H^2 + \eta_t\mathcal{R}^{\ell_1}_{\lambda^{\{K\}}}\left(\theta^{(t)} + \Delta\theta^{(t)}\right) + (1 - \eta_t)\mathcal{R}^{\ell_1}_{\lambda^{\{K\}}}(\theta^{(t)})$$
$$\geq \nabla\mathcal{L}\left(\theta^{(t)}\right)^\top \cdot \Delta\theta^{(t)} + \frac{1}{2}\left\|\Delta\theta^{(t)}\right\|_H^2 + \mathcal{R}^{\ell_1}_{\lambda^{\{K\}}}\left(\theta^{(t)} + \Delta\theta^{(t)}\right).$$

Rearranging the terms, we obtain

$$(1 - \eta_t)\left(\nabla\mathcal{L}\left(\theta^{(t)}\right)^\top \cdot \Delta\theta^{(t)} + \mathcal{R}^{\ell_1}_{\lambda^{\{K\}}}\left(\theta^{(t)} - \Delta\theta^{(t)}\right) - \mathcal{R}^{\ell_1}_{\lambda^{\{K\}}}(\theta^{(t)})\right) + \frac{1 - \eta_t^2}{2}\left\|\Delta\theta^{(t)}\right\|_H^2$$
$$\leq 0$$

Canceling the $(1 - \eta_t)$ factor from both sides and let $\eta_t \to 1$, we obtain the desired inequality,

$$\gamma_t \leq -\left\|\Delta\theta^{(t)}\right\|_H^2.$$

## D.5  Proof of Lemma 16

We first demonstrate an upper bound of the approximate KKT parameter $\omega_{\lambda^{\{K\}}}$. Given the solution $\theta^{(t-1)}$ from the $(t-1)$-th iteration, the optimal solution at $t$-th iteration satisfies the KKT condition:
$$\nabla^2 \mathcal{L}(\theta^{(t-1)})(\theta^{(t)} - \theta^{(t-1)}) + \nabla \mathcal{L}(\theta^{(t-1)}) + \lambda^{\{K\}} \odot \xi^{(t)} = 0,$$
where $\xi^{(t)} \in \partial ||\theta^{(t)}||_1$. Then for any vector $v$ with $||v||_2 \le ||v||_1 = 1$ and $||v||_0 \le s^* + 2\widetilde{s}$, we have
$$
\begin{aligned}
(\nabla \mathcal{L}(\theta^{(t)}) + \lambda^{\{K\}} \odot \xi^{(t)})^\top v &= (\nabla \mathcal{L}(\theta^{(t)}))^\top v - (\nabla^2 \mathcal{L}(\theta^{(t-1)})(\theta^{(t)} - \theta^{(t-1)}) + \nabla \mathcal{L}(\theta^{(t-1)}))^\top v \\
&= (\nabla \mathcal{L}(\theta^{(t)}) - \nabla \mathcal{L}(\theta^{(t-1)}))^\top v - (\nabla^2 \mathcal{L}(\theta^{(t-1)})(\theta^{(t)} - \theta^{(t-1)}))^\top v \\
&\overset{(i)}{\le} \left\| (\nabla^2 \mathcal{L}(\widetilde{\theta}))^{\frac{1}{2}}(\theta^{(t)} - \theta^{(t-1)}) \right\|_2 \cdot \left\| v^\top (\nabla^2 \mathcal{L}(\widetilde{\theta}))^{\frac{1}{2}} \right\|_2 \\
&\quad + \left\| (\nabla^2 \mathcal{L}(\theta^{(t-1)}))^{\frac{1}{2}}(\theta^{(t)} - \theta^{(t-1)}) \right\|_2 \cdot \left\| v^\top (\nabla^2 \mathcal{L}(\theta^{(t-1)}))^{\frac{1}{2}} \right\|_2 \\
&\overset{(ii)}{\le} 2\rho^+_{s^*+2\widetilde{s}} \left\| \theta^{(t)} - \theta^{(t-1)} \right\|_2, \tag{21}
\end{aligned}
$$
where $(i)$ is from mean value theorem with some $\widetilde{\theta} = (1-a)\theta^{(t-1)} + a\theta^{(t)}$ for some $a \in [0,1]$ and Cauchy-Schwarz inequality, and $(ii)$ is from the SE properties. Take the supremum of the L.H.S. of (21) with respect to $v$, we have
$$\left\| \nabla \mathcal{L}(\theta^{(t)}) + \lambda^{\{K\}} \odot \xi^{(t)} \right\|_\infty \le 2\rho^+_{s^*+2\widetilde{s}} \left\| \theta^{(t)} - \theta^{(t-1)} \right\|_2. \tag{22}$$
Then from Lemma 12, we have
$$\left\| \theta^{(t+1)} - \overline{\theta}^{\{K\}} \right\|_2 \le \left( \frac{L_{s^*+2\widetilde{s}}}{2\rho^-_{s^*+2\widetilde{s}}} \right)^{1+2+4+\dots+2^{t-1}} \left\| \theta^{(0)} - \overline{\theta}^{\{K\}} \right\|_2^{2^\top} \le \left( \frac{L_{s^*+2\widetilde{s}}}{2\rho^-_{s^*+2\widetilde{s}}} \left\| \theta^{(0)} - \overline{\theta}^{\{K\}} \right\|_2 \right)^{2^t}.$$
By (22) and (20) by taking $\Delta\theta^{(t-1)} = \theta^{(t)} - \theta^{(t-1)}$, we obtain
$$
\begin{aligned}
\omega_{\lambda^{\{K\}}}\left( \theta^{(t)} \right) &\le 2\rho^+_{s^*+2\widetilde{s}} \left\| \theta^{(t)} - \theta^{(t-1)} \right\|_2 \le 3\rho^+_{s^*+2\widetilde{s}} \left\| \theta^{(t-1)} - \overline{\theta}^{\{K\}} \right\|_2 \\
&\le 3\rho^+_{s^*+2\widetilde{s}} \left( \frac{L_{s^*+2\widetilde{s}}}{2\rho^-_{s^*+2\widetilde{s}}} \left\| \theta^{(0)} - \overline{\theta}^{\{K\}} \right\|_2 \right)^{2^t}.
\end{aligned}
$$
By requiring the R.H.S. equal to $\varepsilon$ we obtain
$$
\begin{aligned}
t &= \log \frac{\log\left( \frac{3\rho^+_{s^*+2\widetilde{s}}}{\varepsilon} \right)}{\log\left( \frac{2\rho^-_{s^*+2\widetilde{s}}}{L_{s^*+2\widetilde{s}} \| \theta^{(0)} - \overline{\theta}^{\{K\}} \|_2} \right)} = \log\log\left( \frac{3\rho^+_{s^*+2\widetilde{s}}}{\varepsilon} \right) - \log\log\left( \frac{2\rho^-_{s^*+2\widetilde{s}}}{L_{s^*+2\widetilde{s}} \| \theta^{(0)} - \overline{\theta}^{\{K\}} \|_2} \right) \\
&\overset{(i)}{\le} \log\log\left( \frac{3\rho^+_{s^*+2\widetilde{s}}}{\varepsilon} \right) - \log\log 4 \le \log\log\left( \frac{3\rho^+_{s^*+2\widetilde{s}}}{\varepsilon} \right),
\end{aligned}
$$
where $(i)$ is from the fact that $\left\| \theta^{(0)} - \overline{\theta}^{\{K\}} \right\|_2 \le R = \frac{\rho^-_{s^*+2\widetilde{s}}}{2L_{s^*+2\widetilde{s}}}$.

# E  Proof of Intermediate Results for Theorem 4

## E.1  Proof of Lemma 8

Given the assumptions, we will show that for all large enough $t$, we have
$$||\theta^{(t+1)}_{\mathcal{S}_\perp}||_0 \le \widetilde{s}.$$
Following the analysis of Lemma 14, Lemma 15, and Appendix H, we have that the objective $\mathcal{F}_{\lambda^{\{1\}}}$ has sufficient descendant in each iteration of proximal Newton step, which is also discussed in [37]. Then there exists a constant $T$ such that for all $t \ge T$, we have
$$\mathcal{F}_{\lambda^{\{1\}}}(\theta^{(t)}) \le \mathcal{F}_{\lambda^{\{1\}}}(\theta^*) + \frac{\lambda_{\text{tgt}}}{4}||\theta^{(t)} - \theta^*||_1,$$
where $||\theta^{(t)} - \theta^*||_1 \le c\lambda_{\text{tgt}}\sqrt{s^*}/\rho^-_{s^*+\widetilde{s}}$ from similar analysis in [9]. The rest of the analysis is analogous to that of Lemma 11, from which we have $||\theta^{(t)}_{\mathcal{S}_\perp}||_0 \le \widetilde{s}$.

### E.2 Proof of Lemma 9

The estimation error is derived analogously from [9], thus we omit it here. The claim of the quadratic convergence follows directly from Lemma 12 given sparse solutions.

### E.3 Proof of Lemma 10

The upper bound of the number of iterations for proximal Newton update is obtained by combining Lemma 8 and Lemma 16. Note that

$$
T_1 \leq \log \frac{\log \left( \frac{3\rho^+_{s^*+2\widetilde{s}}}{\varepsilon} \right)}{\log \left( \frac{2\rho^-_{s^*+2\widetilde{s}}}{L_{s^*+2\widetilde{s}} \left\| \theta^{(T+1)} - \overline{\theta}^{\{1\}} \right\|_2} \right)}.
$$

Then we obtain the result from $\left\| \theta^{(T+1)} - \overline{\theta}^{\{1\}} \right\|_2 \leq R = \frac{\rho^-_{s^*+2\widetilde{s}}}{2L_{s^*+2\widetilde{s}}}$.

## F   Proof of Theorem 7

It is demonstrated in [24] that Assumptions $1 \sim 3$ hold given the LRE properties defined in Definition 18. Thus, combining the analyses in [24] and Proposition 19, we have that Assumptions $1 \sim 3$ hold with high probability. Assumption 4 also holds trivially by choosing $\varepsilon = \frac{c}{\sqrt{n}}$ for some generic constant $c$. The rest of the results follow directly from Theorem 5 and the analyses in [40].

## G   Further Intermediate Results

**Lemma 20.** *Given* $\omega_{\lambda^{\{K\}}}(\widehat{\theta}^{\{K\}}) \leq \frac{\lambda_{\mathrm{tgt}}}{8}$, *we have that for all* $t \geq 1$ *at the* $\{K+1\}$-*th stage,*

$$
\omega_{\lambda^{\{K+1\}}}(\theta^{(t)}) \leq \frac{\lambda_{\mathrm{tgt}}}{4} \quad \text{and} \quad \mathcal{F}_{\lambda^{\{K+1\}}}(\theta^{(t)}) \leq \mathcal{F}_{\lambda^{\{K+1\}}}(\theta^*) + \frac{\lambda_{\mathrm{tgt}}}{4}||\theta^{(t)} - \theta^*||_1.
$$

*Proof.* Note that at the $\{K+1\}$-th stage, $\theta^{(0)} = \widehat{\theta}^{\{K\}}$. Then we have

$$
\omega_{\lambda^{\{K+1\}}}(\theta^{(0)}) = \min_{\xi \in \partial||\theta^{(0)}||_1} ||\nabla\mathcal{L}(\theta^{(0)}) + \lambda^{\{K+1\}} \odot \xi||_\infty
$$

$$
\overset{(i)}{\leq} \min_{\xi \in ||\theta^{(0)}||_1} ||\nabla\mathcal{L}(\theta^{(0)}) + \lambda^{\{K\}} \odot \xi||_\infty + ||(\lambda^{\{K+1\}} - \lambda^{\{K\}}) \odot \xi||_\infty
$$

$$
\overset{(ii)}{\leq} \omega_{\lambda^{\{K\}}}(\theta^{(0)}) + ||\lambda^{\{K+1\}} - \lambda^{\{K\}}||_\infty \overset{(iii)}{\leq} \frac{\lambda_{\mathrm{tgt}}}{8} + \frac{\lambda_{\mathrm{tgt}}}{8} \leq \frac{\lambda_{\mathrm{tgt}}}{4},
$$

where $(i)$ is from triangle inequality, $(ii)$ is from the definition of the approximate KKT condition and $\xi$, and $(iii)$ is from $\omega_{\lambda^{\{K\}}}(\theta^{(0)}) = \omega_{\lambda^{\{K\}}}(\widehat{\theta}^{\{K\}}) \leq \frac{\lambda_{\mathrm{tgt}}}{8}$ and $||\lambda^{\{K+1\}} - \lambda^{\{K\}}||_\infty \leq \frac{\lambda_{\mathrm{tgt}}}{8}$.

For some $\xi^{(t)} = \mathrm{argmin}_{\xi \in \partial||\theta^{(t)}||_1} ||\nabla\mathcal{L}(\theta^{(t)}) + \lambda^{\{K+1\}} \odot \xi||_\infty$, we have

$$
\mathcal{F}_{\lambda^{\{K+1\}}}(\theta^*) \overset{(i)}{\geq} \mathcal{F}_{\lambda^{\{K+1\}}}(\theta^{(t)}) - (\nabla\mathcal{L}(\theta^{(t)}) + \lambda^{\{K+1\}} \odot \xi^{(t)})^\top (\theta^{(t)} - \theta^*)
$$

$$
\geq \mathcal{F}_{\lambda^{\{K+1\}}}(\theta^{(t)}) - ||\nabla\mathcal{L}(\theta^{(t)}) + \lambda^{\{K+1\}} \odot \xi^{(t)}||_\infty ||\theta^{(t)} - \theta^*||_1
$$

$$
\overset{(ii)}{\geq} \mathcal{F}_{\lambda^{\{K+1\}}}(\theta^{(t)}) - \frac{\lambda_{\mathrm{tgt}}}{4}||\theta^{(t)} - \theta^*||_1
$$

where $(i)$ is from the convexity of $\mathcal{F}_{\lambda^{\{K+1\}}}$ and $(ii)$ is from the fact that for all $t \geq 0$, $||\nabla\mathcal{L}(\theta^{(t)}) + \lambda^{\{K+1\}} \odot \xi^{(t)}||_\infty \leq \frac{\lambda_{\mathrm{tgt}}}{4}$. This finishes the proof. $\qquad\square$

**Lemma 21** (Adapted from [9]). *Suppose* $||\theta^{(t)}_{\mathcal{S}_\perp}||_0 \leq \widetilde{s}$ *and* $\omega_{\lambda^{\{K\}}}(\theta^{(t)}) \leq \frac{\lambda_{\mathrm{tgt}}}{4}$. *Then there exists a generic constant* $c_1$ *such that*

$$
||\theta^{(t)} - \theta^*||_2 \leq \frac{c_1\lambda_{\mathrm{tgt}}\sqrt{s^*}}{\rho^-_{s^*+2\widetilde{s}}}.
$$

# H Global Convergence Analysis

For notational convenience, we denote $\mathcal{F} = \mathcal{F}_\lambda$ and $\mathcal{R} = \mathcal{R}_\lambda^{\ell_1}$ in the sequel. We first provide an upper bound of the objective gap.

**Lemma 22.** *Suppose the $\mathcal{F}(\theta) = \mathcal{R}(\theta) + \mathcal{L}(\theta)$ and $\mathcal{L}(\theta)$ satisfies the restricted Hessian smoothness property, namely, for any $\theta, h \in \mathbb{R}^d$*

$$\frac{d}{d\tau}\nabla^2\mathcal{L}(\theta + \tau h)|_{\tau=0} \preceq C\sqrt{h^\top \nabla^2\mathcal{L}(\theta)h} \cdot \nabla^2\mathcal{L}(\theta),$$

*for some constant $C$. Let $\Delta\theta$ be the search direction and let $\theta_+ = \theta + \tau\Delta\theta$ for some $\tau \in (0, 1]$. Then*

$$\mathcal{F}(\theta_+) \leq \mathcal{F}(\theta) + \left[-\tau + \mathcal{O}(\tau^2)\right]\|\Delta\theta\|_H^2.$$

*Proof.* From the convexity of $\mathcal{R}$, we have

$$\begin{aligned}
\mathcal{F}(\theta_+) - \mathcal{F}(\theta) &= \mathcal{L}(\theta_+) - \mathcal{L}(\theta) + \mathcal{R}(\theta_+) - \mathcal{R}(\theta)\\
&\leq \mathcal{L}(\theta_+) - \mathcal{L}(\theta) + \tau\mathcal{R}(\theta + \Delta\theta) + (1-\tau)\mathcal{R}(\theta) - \mathcal{R}(\theta)\\
&= \mathcal{L}(\theta_+) - \mathcal{L}(\theta) + \tau\left(\mathcal{R}(\theta + \Delta\theta) - \mathcal{R}(\theta)\right)\\
&= \nabla\mathcal{L}(\theta)^\top \cdot (\tau\Delta\theta) + \tau\left(\mathcal{R}(\theta + \Delta\theta) - \mathcal{R}(\theta)\right) + \tau\int_0^\tau (\Delta\theta)^\top\nabla^2\mathcal{L}(\theta + \alpha\Delta\theta)\Delta\theta\, d\alpha.
\end{aligned}$$

By Lemma 15 and the restricted Hessian smoothness property, we obtain

$$\begin{aligned}
&\mathcal{F}(\theta_+) - \mathcal{F}(\theta)\\
&\leq -\tau\|\Delta\theta\|_{\nabla^2\mathcal{L}(\theta)} + \tau\int_0^\tau (\Delta\theta)^\top\nabla^2\mathcal{L}(\theta + \alpha\Delta\theta)\Delta\theta\, d\alpha\\
&= -\tau\|\Delta\theta\|_{\nabla^2\mathcal{L}(\theta)} + \tau\int_0^\tau d\alpha\int_0^\alpha dz\frac{d}{dz}(\Delta\theta)^\top\nabla^2\mathcal{L}(\theta + z\Delta\theta)\Delta\theta + \tau\int_0^\tau d\alpha(\Delta\theta)^\top\nabla^2\mathcal{L}(\theta)\Delta\theta\\
&= \left(-\tau + \mathcal{O}(\tau^2)\right)\|\Delta\theta\|_{\nabla^2\mathcal{L}(\theta)}^2.
\end{aligned}$$

$\square$

Next, we show that $\Delta\theta \neq 0$ when $\theta$ have not attained the optimum.

**Lemma 23.** *Suppose the $\mathcal{F}(\theta) = \mathcal{R}(\theta) + \mathcal{L}(\theta)$ has a unique minimizer, and $\mathcal{L}(\theta)$ satisfies the restricted Hessian smoothness property. Then $\Delta\theta^{(t)} = 0$ if and only if $\theta^{(t)} = \bar{\theta}$.*

*Proof.* Suppose $\Delta\theta$ is non-zero at $\bar{\theta}$. Lemma 22 implies that for sufficiently small $0 < \tau \leq 1$,

$$\mathcal{F}(\bar{\theta} + \tau\Delta\theta^{(t)}) - \mathcal{F}(\bar{\theta}) \leq 0.$$

However $\mathcal{F}(\theta)$ is uniquely minimized at $\bar{\theta}$, which is a contradiction. Thus $\Delta\theta = 0$ at $\bar{\theta}$.

Now we consider the other direction. Suppose $\Delta\theta = 0$, then $\theta$ is a minimizer of $\mathcal{F}$. Thus for any direction $h$ and $\tau \in (0, 1]$, we obtain

$$\nabla\mathcal{L}(\theta)^\top(\tau h) + \frac{1}{2}\tau^2 h^\top Hh + \mathcal{R}(\theta + \tau h) - \mathcal{R}(\theta) \geq 0.$$

Rearrange, we obtain

$$\mathcal{R}(\theta + \tau h) - \mathcal{R}(\theta) \geq -\tau\nabla\mathcal{L}(\theta)^\top h - \frac{1}{2}\tau^2 h^\top Hh$$

Let $D\mathcal{F}(\theta, h)$ be the directional derivative of $\mathcal{F}$ at $\theta$ in the direction $h$, thus

$$\begin{aligned}
D\mathcal{F}(\theta, h) &= \lim_{\tau\to 0}\frac{\mathcal{F}(\theta + \tau h) - \mathcal{F}(\theta)}{\tau}\\
&= \lim_{\tau\to 0}\frac{\tau\nabla\mathcal{L}(\theta)^\top h + \mathcal{O}(\tau^2) + \mathcal{R}(\theta + \tau h) - \mathcal{R}(\theta)}{\tau}\\
&\geq \lim_{\tau\to 0}\frac{\tau\nabla\mathcal{L}(\theta)^\top h + \mathcal{O}(\tau^2) - \tau\nabla\mathcal{L}(\theta)^\top h - \frac{1}{2}\tau^2 h^\top Hh}{\tau} = 0.
\end{aligned}$$

Since $\mathcal{F}$ is convex, then $\theta$ is the minimizer of $\mathcal{F}$. $\square$

Then, we show the behavior of $\|\Delta\theta\|_H$ and $\mathcal{R}(\theta + \Delta\theta)$ when $\Delta\theta \neq 0$.

**Lemma 24.** *Suppose at any point $\theta \in \mathbb{R}^d$, we have $\nabla\mathcal{L}(\theta) \in \text{span}\left(\nabla^2\mathcal{L}(\theta)\right)$. If $\Delta\theta \neq 0$ then either*

$$\|\Delta\theta\|_H > 0 \quad or \quad \mathcal{R}\left(\theta + \Delta\theta\right) < \mathcal{R}\left(\theta\right).$$

*Proof.* Recall that $\Delta\theta$ is obtained by solving the following sub-problem,

$$\Delta\theta = \underset{\Delta\theta}{\text{argmin}}\, \mathcal{R}(\theta + \Delta\theta) + \nabla\mathcal{L}(\theta)^\top \Delta\theta + \|\Delta\theta\|_H^2.$$

If $\|\Delta\theta\|_H = 0$ and $\Delta\theta \neq 0$, then

$$\Delta\theta \perp \text{span}(H) \quad \text{and} \quad \nabla\mathcal{L}(\theta)^\top \Delta\theta = 0.$$

Thus

$$\mathcal{R}\left(\theta + \Delta\theta\right) < \mathcal{R}\left(\theta\right).$$

Notice that $\mathcal{R}\left(\theta + \Delta\theta\right) \neq \mathcal{R}\left(\theta\right)$, since otherwise $\Delta\theta = 0$ is a solution. $\quad\square$

Finally, we demonstrate the strict decrease of the objective in each proximal Newton step.

**Lemma 25.** *Suppose at any point $\theta \in \mathbb{R}^d$, we have $\nabla\mathcal{L}(\theta) \in \text{span}\left(\nabla^2\mathcal{L}(\theta)\right)$. If $\Delta\theta \neq 0$ then*

$$\mathcal{F}(\theta + \tau\Delta\theta) < \mathcal{F}(\theta),$$

*for small enough $\tau > 0$.*

*Proof.* By Lemma 24, if $\Delta\theta \neq 0$, then either $\|\Delta\theta\|_H > 0$ or $\mathcal{R}(\theta + \Delta\theta) - \mathcal{R}(\theta) < 0$. If it is the first case, then by Lemma 15,

$$\gamma = \nabla\mathcal{L}(\theta)^\top \Delta\theta + \mathcal{R}(\theta + \Delta\theta) - \mathcal{R}(\theta) < -\|\Delta\theta\|_H < 0.$$

It is the second case, then $\nabla\mathcal{L}(\theta)^\top \Delta\theta = 0$ and

$$\gamma = \mathcal{R}(\theta + \Delta\theta) - \mathcal{R}(\theta) < 0.$$

Moreover, we have

$$\begin{aligned}
&\mathcal{F}(\theta + \tau\Delta\theta) - \mathcal{F}(\theta) \\
&= \mathcal{L}(\theta + \tau\Delta\theta) - \mathcal{L}(\theta) + \mathcal{R}(\theta + \tau\Delta\theta) - \mathcal{R}(\theta) \\
&\leq \tau\nabla\mathcal{L}(\theta)^\top \Delta\theta + \frac{\tau^2}{2}\Delta\theta^\top H\Delta\theta + \mathcal{O}(\tau^3) + \mathcal{R}(\theta + \tau\Delta\theta) - \mathcal{R}(\theta) \\
&\leq \tau\nabla\mathcal{L}(\theta)^\top \Delta\theta + \tau\mathcal{R}(\theta + \Delta\theta) + (1 - \tau)\mathcal{R}(\theta) - \mathcal{R}(\theta) + \frac{\tau^2}{2}\Delta\theta^\top H\Delta\theta + \mathcal{O}(\tau^3) \\
&= \tau(\gamma + \mathcal{O}(\tau)).
\end{aligned}$$

where the first inequality is from the restricted Hessian smoothness property. Thus $\mathcal{F}(\theta + \tau\Delta\theta) - \mathcal{F}(\theta) < 0$ for sufficiently small $\tau > 0$. $\quad\square$

Since each step, the objective is strictly decreasing, thus the algorithm will eventually reach the minimum.