[Reviews · NeurIPS 2017]

Reviewer 1



This paper discusses a difference of convex proximal Newton algorithm for solving sparse non-convex problems.The authors claim that this is the first work in second order approaches for high-dimensional sparse learning for both convex and non-convex regularizers. They also provide statistical and computational guarantees. However, the paper has certain weaknesses that are addressed below: * Most of the theoretical work presented here are built upon prior work, it is not clear what is the novelty and research contribution of the paper. * The figures are small and almost unreadable * It doesn't clearly state how equation 5, follows from equation 4 * It is not clear how \theta^{t+1/2} come into the picture. Explain * S^{*} and S^{~} are very important parameters in the paper, yet they were not properly defined. In line 163 it is claimed that S^{~} is defined in Assumption 1, but one can see the definition is not proper, it is rather cyclic. * Since the comparison metric here is wall-clock time, it is imperative that the implementation of the algorithms be the same. It is not clear that it is guaranteed. Also, the size of the experimental data is quite small. * If we look into the run-times of DCPN for sim_1k, sim_5k, and sim_10k and compare with DC+ACD, we see that DCPN is performing better, which is good. But the trend line tells us a different story; between 1k and 10k data the DCPN run-time is about 8x while the competitor grows by only 2x. From this trend it looks like the proposed algorithm will perform inferior to the competitor when the data size is larger e.g., 100K. * Typo in line 106

Reviewer 2



Summary: This paper considers a DC (difference of convex) approach to solve *sparse* ERM problems with a non-convex regularization (in high dimensions). The specific non-convex regularizer studied in this paper is the capped l1 penalty. Now, given a particular convex relaxation via the DC approach, this paper proposes to solve the problem using proximal Newton. The paper analyses the l2 convergence of their solution (at any step of the convex relaxation) to the truth, and shows that it enjoys quadratic convergence to stastical rates. The analysis is contingent on a Sparse Eigenvalue (similar to RSC) condition and a restricted Hessian smoothness condition. Comments: Overall, this is a well written paper and the results presented seem sound. I do however have the following questions/comments: 1. What is T in the warm initialization step of the proposed algorithm ? What does it scale as i.e. what is its dependence on other constants involved in the proof ? 2. It would be good if the authors could provide a more detailed comparison with [36] in the related work. It seems that [36] also considers proximal Newton. However, [36] only considers a convex regularizer. Is another consequence of this work also that the analysis of [36] can be improved with weaker conditions of this paper ? 3. Theorem 6 shows that for large enough K, and small s' (i.e. none or few weak signals), the algo in this paper can yield a l2-rate of O(\sqrt{s*/n}). However, if I am not wrong, there is also a minimax lower bound of O(\sqrt{s* log d / n}) i.e. an extra log d factor (for example, see Theorem 1(b) in https://arxiv.org/abs/0910.2042). Am I missing something here ? How can these two rates be reconciled ? 4. In the experiments, the authors should specify how \beta (the tuning parameter for capped l1) was chosen. EDIT: I have seen the authors' response, and my rating stands.

Reviewer 3



The authors present a DC proximal Newton algorithm for nonconvex regularized sparse learning problems in high dimensions. The method is shown to obtain local quadratic convergence at each stage of convex relaxation while maintaining solution sparsity by exploiting characteristic structures of sparse modelling (i.e., restricted strong convexity and Hessian smoothness). This paper is very clear and I find it very interesting. To the best of my knowledge, the details of the analysis are sound and this is a significant contribution towards our understanding of how second-order methods in high dimensions have superior performance (both empirically and computationally) despite the lack of theoretical analysis in the past for nonconvex regularized sparse modelling approaches. Questions: - In the numerical results, how much of the time in the Table 1 results was the Warm Initialization step? As I understand, Figure 5 shows the convex relaxation stages after warm initialization. Is that correct? Do the approaches the authors compare against (DC + APG and DC + ACD) also use a warm initialization step? - Can the authors comment on if there are non-additive L(\theta) functions that might be of interest and if so, how the analysis in this work might extend to this setting? ============== POST REBUTTAL: ============== I have read the author rebuttal and the other reviews. I thank the authors for addressing my comments and questions. My recommendation is still for acceptance.